

# Carbon sources of benthic fauna in temperate lakes across multiple trophic states

Annika Fiskal[1], Eva Anthamatten[1], Longhui Deng[1], Xingguo Han[1], Lorenzo Lagostina[1], Anja Michel[1], Rong Zhu[1], Nathalie Dubois[2,3], Carsten J. Schubert[1,4], Stefano M. Bernasconi[3], Mark A. Lever[1]

[1]Institute of Biogeochemistry and Pollutant Dynamics (IBP), ETH Zurich, Universitätstrasse 16, 8092 Zurich, Switzerland
[2]Surface Waters Research – Management, Eawag, Swiss Federal Institute of Aquatic Science and Technology, Überlandstrasse 133, 8600 Dübendorf, Switzerland
[3]Department of Earth Sciences, ETH Zurich, Sonneggstrasse 5, 8092 Zurich, Switzerland
[4]Department of Surface Waters – Research and Management, Swiss Federal Institute of Aquatic Science and Technology (EAWAG), Seestrasse 79, 6047 Kastanienbaum, Switzerland

Correspondence: Annika Fiskal (annikafiskal@gmail.com) and Mark A. Lever (mark.lever@usys.ethz.ch)

**Keywords**

methane-derived carbon, benthic macrofauna, lacustrine sediment, oligochaetes, chironomid larvae, methane-oxidizing bacteria, endosymbionts

**Abstract.** Previous studies have shown that microbially produced methane can be a dominant carbon source of lacustrine sedimentary macrofauna in eutrophic lakes, most likely through grazing on methane-oxidizing bacteria. Here we investigate the contributions of different carbon sources to macrofaunal biomass across five lakes in central Switzerland that range from oligotrophic to highly eutrophic. Macrofaunal communities change with trophic state, with chironomid larvae dominating oligotrophic and tubificid oligochaetes dominating eutrophic lake sediments. $^{13}$C-isotopic data suggest that the average contribution of methane-derived carbon to the biomass of both macrofaunal groups is similar, but consistently remains minor, ranging from only ~1% in the oligotrophic lake to at most 12% in the eutrophic lakes. The remaining biomass can be explained with assimilation of detritus-derived organic carbon. Low abundances of methane cycling microorganisms in macrofaunal specimens, burrows, and surrounding sediment based on 16S ribosomal RNA (rRNA) gene sequences and copy numbers of genes involved in anaerobic and aerobic methane cycling (*mcrA*, *pmoA*) support the interpretation of isotopic data. Notably, 16S rRNA gene sequences of macrofauna, including macrofaunal guts, are highly divergent from those in tubes or sediments. Many macrofaunal specimens are dominated by a single 16S rRNA phylotype of *Fusobacteria, α-, β-, γ-,* or *ε-Proteobacteria*, *Bacteroidetes*, or *Parcubacteria*. This raises the question whether dominant lake macrofauna live in so far uncharacterized relationships with detrital organic matter-degrading bacterial endosymbionts.





## 1 Introduction

Lake sediments are globally important organic C sinks (Einsele et al., 2001; Mendonça et al., 2017) and sources of the greenhouse gas methane ($CH_4$) (Bastviken et al., 2004; Raymond et al., 2013; Holgerson and Raymond, 2016). Overall the burial of organic carbon is usually higher in eutrophic compared to oligotrophic lakes due to high nutrient loads which increase primary production (Dean and Gorham, 1998; Maerki et al., 2009; Heathcote and Downing, 2012; Anderson et al., 2013; Anderson et al., 2014). Resulting increases in aerobic respiration lead to $O_2$ depletion and increased organic matter deposition to sediments (Hollander et al., 1992; Steinsberger et al., 2017), where this increased organic matter stimulates microbial methane production (Fiskal et al., 2019). The combination of increased methane production in sediments and decreased aerobic methane consumption in overlying water then results in higher methane emissions from eutrophic lakes (DelSontro et al., 2016).

In addition to trophic state, the presence of macrofauna, which physically mix sediments, mechanically break down organic particles, or pump $O_2$ into deeper, otherwise anoxic layers, influences $O_2$ and C cycle dynamics in sediments (Meysman et al., 2006; White and Miller, 2008; Kristensen et al., 2012). While most research on macrofaunal effects on organic carbon burial and respiration reactions have been on marine sediments, there have also been numerous studies on freshwater sediments. These studies suggest that macrofauna can be present at high abundances (up to 11,000 individuals m$^{-2}$) (Armitage et al., 1995; Mousavi, 2002) and significantly influence nutrient fluxes and sedimentary matrices in lake sediments (Stief, 2013; Holker et al., 2015). Insects, in particular tube-dwelling chironomid larvae, can cause oxic-anoxic oscillations around their burrows through their pumping activity (Lewandowski et al., 2007; Roskosch et al., 2012; Baranov et al., 2016; Hupfer et al., 2019)These redox fluctuations affect the sedimentary cycles of nitrogen (Pelegri et al., 1994; Jeppesen et al., 1998; Stief et al., 2009; Stief, 2013), phosphorus (Andersson et al., 1988; Katsev et al., 2006), iron (Hupfer and Lewandowski, 2008) and methane (Deines et al., 2007b; Gentzel et al., 2012). Worms, especially tubificid oligochaetes, can also increase oxygenation and $O_2$ uptake in (Lagauzère et al., 2009) and influence the release of ammonium ($NH_4^+$), nitrate ($NO_3^-$), and phosphate ($PO_4^{3-}$) (Svensson et al., 2001; Mermillod-Blondin et al., 2005; Gautreau et al., 2020) from surface sediments. Many tubificids are moreover head-down deposit feeders that defecate on the sediment surface (McCall and Tevesz, 1982). This upward movement of reduced sediment can cause significant reworking and alter the redox potential in surface sediment (Davis, 1974).

The community composition of lacustrine sedimentary macrofauna varies in response to trophic state (Aston, 1973; Verdonschot, 1992; Nicacio and Juen, 2015), in part due to differences in hypoxia/anoxia tolerance among macrofaunal species (Chapman et al., 1982). Different lacustrine macrofaunal species, moreover, vary in their impact on methane cycling in sediments (Bussmann, 2005; Figueiredo-Barros et al., 2009). methane oxidation in surface sediments is often stimulated by chironomid larval $O_2$ input, which enriches populations of methane-oxidizing bacteria in larval tubes and surrounding sediment ("microbial gardening") (Kajan and Frenzel, 1999). As a result, methane-oxidizing bacteria can become an important food source, and in some cases the main C source, of chironomid larvae (Kankaala et al., 2006; Deines et al., 2007a; Jones et al., 2008; Jones and Grey, 2011). High contributions of methane-derived carbon via grazing on methane-oxidizing bacteria are typically found in profundal regions of eutrophic lakes with seasonal stratification and low $O_2$ concentrations (Hershey et al., 2006; Jones and Grey, 2011). Yet, variable isotopic values of chironomid biomass, even within the same location, suggest that diets of chironomid larvae vary greatly (Kiyashko et al., 2001; Reuss et al., 2013). The limited C-isotopic data on tubificid worms suggest that worm C sources also vary from detritus-based to locally or seasonally high contributions of methane-derived carbon (Premke et al., 2010).

Despite these past studies, the conditions under which methane-derived carbon becomes an important C source to chironomid larvae or oligochaetes are not well understood. Furthermore, the main pathways of methane-derived carbon incorporation into macrofaunal biomass, e.g. selective grazing or gardening of methane-oxidizing bacteria, or carbon transfer from methane-oxidizing bacteria gut symbionts, remain unclear. Here we analyse shallow sublittoral to profundal sediments of five temperate lakes in central Switzerland that differ strongly in trophic state and macrofaunal community composition.



We analyse the community structures of chironomid larvae and oligochaetes and compare their C-isotopic compositions to those of total organic C (TOC), dissolved organic C (DOC), and methane to investigate how C sources vary across dominant macrofaunal groups in relation to trophic state and water depth. In addition, we analyse microbial community structure based on 16S rRNA gene sequences and quantify functional genes involved in aerobic and anaerobic methane oxidation in macrofaunal specimens, macrofaunal burrows, and surrounding sediment to elucidate the genetic potential for macrofauna-microbiota associations.

## 2    Material and Methods

### 2.1  Sampling and Site description

Sediment cores were obtained from three different water depths in the oligotrophic Lake Lucerne, the mesotrophic Lake Zurich, and the eutrophic Lake Zug, Lake Baldegg, and Lake Greifen in central Switzerland in June and July 2016 (for map of lakes and sampled stations please see Fiskal et al. (2019)). Sediment cores were taken using gravity cores with 60 cm long liners that had an inner diameter of 150 mm (UWITEC, AT) from boats or motorized platforms. The four sediment cores per station were used for microsensor measurements ($O_2$, pH), porewater sampling using rhizones (0.2 µm pore size, Rhizosphere), analyses of DNA sequences, methane concentrations, TOC content, and physical properties, and macrofaunal community analyses, respectively (for further details see (Fiskal et al., 2019)). Cores for macrofaunal community analyses were extruded and macrofauna collected by sieving sediments from different depth intervals (0-4 cm depth: 1-cm intervals; below 4 cm: 2-cm intervals) through 400 and 200 µm mesh sieves. Three stations (two in Lake Lucerne, one in Lake Baldegg) were revisited in November 2017 and October 2018 to collect additional macrofaunal specimens for DNA analyses.

### 2.2  Macrofaunal abundance and taxonomy

For each depth interval, specimen numbers of oligochaetes and chironomid larvae were counted and carefully picked with tweezers and preserved in 70% ethanol for taxonomic and $^{13}$C-isotopic analyses or frozen on dry ice for DNA extractions. Detailed taxonomic analyses to the genus and, where possible, species level were performed on a subset of oligochaetes and chironomid larvae. Oligochaete specimens were sent to AquaLytis (Wildau, Germany), where they were embedded in epoxy resin and identified by light microscopy. Chironomid larvae were microscopically identified by AquaDiptera (Emmendingen, Germany).

### 2.3 Stable carbon isotope analyses

Carbon isotope analyses were performed on DOC, methane, TOC, macrofaunal specimens, or separately on guts and remaining bodies of macrofaunal specimens. Values are given in the $\delta$ notation; i.e.:

$$\delta^{13}C = [(^{13}C/^{12}C)_{sample}/(^{13}C/^{12}C)_{standard}].$$

*$\delta^{13}$C-DOC.*  Porewater samples were analyzed as described in Lang et al. (2012). Briefly, 2-7 ml of sample were added to 12 ml vacutainers. After removal of dissolved inorganic C by addition of 85% phosphoric acid and bubbling with high purity He, DOC was oxidized to $CO_2$ using persulfate (1h at 100°C). The evolved $CO_2$ was analyzed on a Gasbench II coupled to a Delta V mass spectrometer (Thermo Fisher Scientific, Bremen). Water soluble organic standards of known isotope composition (phthalic acid and sucrose) were used as standards.

*$\delta^{13}$C-Methane.* Methane was extracted by creating a sediment slurry with MilliQ water under saturating NaCl concentrations (~6.3 M). 2 cm$^3$ of sediment were transferred to 20 ml crimp vials containing 2.514g NaCl and 5 ml MilliQ water, crimped, mixed, and stored on ice or at +4 ☐C until analysis using a Trace Gas (Isoprime) coupled to an isotope ratio mass spectrometer (GC-IRMS, Isoprime, Manchester). Separation was performed through a GC-column (PoraPLOT Q 30m column). The





precision of the method was ± 0.7 ‰. After every 6[th] sample we included a standard with a known $\delta^{13}$C value (Standards: L-iso1 with 2,500 ppmv CH4 at -66.5 ‰ $\delta^{13}$C-methane and T-iso3 with 250 ppmv CH4 at -38.3 ‰ $\delta^{13}$C-CH4, Air Liquide).

*$\delta^{13}$C-TOC.* For $\delta^{13}$C-TOC analyses, 5–10 g of frozen sediment were freeze-dried in glass vials and quantified using an Elemental analyzer (Thermo Fisher Flash EA 1112) coupled to an isotope-ratio-mass spectrometer (Thermo Fisher Delta V Plus) (EA-IRMS) as outlined in Fiskal et al. (2019).

*$\delta^{13}$C-Macrofauna.* $\delta^{13}$C-analyses were performed on macrofaunal biomass according to the same method used for $\delta^{13}$C-TOC. Single specimens were cleaned with molecular grade water to remove sediment. Whole organisms, or separated guts and residual bodies, were placed in tin foil capsules, which were mounted to 96-well plates. 96-well plates were sealed using plastic seal foil, the foil above each well was pierced, and the whole plate was freeze dried. Afterward, the foil was removed, the tin foil capsules were closed, and the $\delta^{13}$C of macrofaunal biomass was measured.

## 2.4 Two end member mixing model

Assuming TOC and methane as the only food sources, a two end member mixing model was used to estimate the contribution of methane to biomass C of macrofauna:

$$CH_4\text{-Contribution (\%)} = (100 - (\delta^{13}C_{fauna} - \delta^{13}C_{CH_4})/(\delta^{13}C_{TOC} - \delta^{13}C_{CH_4})*100)$$

## 2.5 DNA extraction from macrofauna

DNA from macrofauna samples was extracted following lysis protocol II in Lever *et al.* (2015). After removal of sediment by rinsing with molecular grade water, DNA was either extracted from entire specimens, or separately on guts and the remaining body. All specimens were cut into 3-4 pieces using a 70% ethanol-cleaned, flame-sterilized scalpel to increase extraction efficiency and then added to 2-mL screw cap tubes filled to ~15% with 0.1 mm Zirconium beads. 50 µl of sodium hexametaphosphate solution were added and each specimen soaked by gentle shaking. Afterwards 0.5 ml of lysis solution I (30 mM Tris-HCl, 30 mM disodium EDTA, 1% Triton X-100, 800 mM guanidium hydrochloride, 2×autoclaved, pH was raised to 10.0 with 5M NaOH) was added and mixed with the sample by inverting and brief vortexing. Subsequently each sample was bead beaten (Labgene, Precellys 24, 5000 rpm, 30 s) and incubated for 1 h at 50 °C on an Eppendorf Thermomixer set to 600 rpm. Samples were then centrifuged (10,000 g, 4°C, 10 min), the transferred supernatants washed twice with ice-cold chloroform-isoamyl alcohol (24:1), and DNA precipitated at room temperature for 2 h in the dark in a linear polyacrylamide-sodium chloride-ethanol solution. The resulting DNA pellets were dissolved in molecular grade water and purified using the CleanAll DNA/RNA Clean-Up and Concentration Micro Kit by Norgen Biotek (protocol A). All Norgen kit chemicals and tubes were autoclaved for 8 h before use to eliminate background DNA contamination. For further details on the extraction method, see Lever et al. (2015). All DNA extracts were stored at -80 °C until further analyses.

## 2.6 qPCR

Quantitative polymerase chain reaction (qPCR) was performed to quantify bacterial and archaeal 16S rRNA genes, and genes encoding particulate methane monooxygenase (*pmoA*) and methyl coenzyme M reductase (*mcrA*) (Table 1). Standards consisting of plasmids containing 16S rRNA, *pmoA*, or *mcrA* genes from specific organisms (Table 1) were run in 10fold dilutions of ~10[1] to ~10[7] gene copies per qPCR reaction. All sample DNA extracts and standard dilutions were run in duplicate.

The qPCR protocols are shown in Table 2. For each qPCR reaction, 2 µl of DNA extract were mixed with 1 µl of molecular grade water, 1 µl of bovine serum albumin (10 mg/ml; New England Biolabs, USA), 0.5 µl each forward and reverse primers (10 µM), and 5 µl LightCycler 480 SYBR Green I Master Mix (Roche, Switzerland). All standards and samples were kept on ice throughout the preparations and run immediately after in transparent 96-well plates on a Roche LightCycler 480.

2.7 Next Generation Sequencing (NGS) and Bioinformatic analyses



Libraries of bacterial and archaeal communities were produced using the universal 16S rRNA primer pair Univ519F and Univ802R (Claesson et al., 2009; Wang and Qian, 2009). Library preparations and subsequent data processing, including 97% zero-radius operational taxonomic unit (ZOTU) clustering, were done as outlined in Han et al. (2020; for PCR reaction mixtures and cycler conditions see Table S1). Principal Coordinates Analysis (PCoA) on bacterial communities at the phylum,
class, order, family, and genus level were performed using Bray-Curtis distances in R (Team, 2018).

## 2.8 Statistical analyses

Statistical differences between C isotope signatures of macrofauna and C pools (methane, TOC) from the same depth interval, and of percentages of bacterial 16S rRNA, *mcrA* and *pmoA* gene copy numbers relative to total 16S rRNA gene copy numbers across sample types (macrofauna, larval tube, sediment) from the same depth interval were determined using Wilcoxon Sign
Rank Tests for paired data. All tests were performed in R (Team, 2018) using the command: wilcox.test (A, B, paired = TRUE, alternative = "two.sided" for (a), "greater/less" for (b), mu = 0.0, exact = TRUE, correct = TRUE, conf.int = TRUE, conf.level = 0.95).

## 3     Results

### 3.1 Macrofaunal distribution in relation to lake trophic state

Macrofauna is present at all stations except the hypoxic deep station of Lake Zurich and is dominated by oligochaetes and chironomid larvae. While oligochaetes are present in all lakes, no chironomid larvae were found in Lake Greifen. Oligochaete densities increase with trophic state, from $75\pm86$ m$^{-2}$ in Lake Lucerne to $4849\pm4443$ m$^{-2}$ in Lake Baldegg. Numbers of chironomid larvae show the opposite trend, decreasing from $641\pm346$ m$^{-2}$ in Lake Lucerne and $849\pm160$ m$^{-2}$ in Lake Zurich to less than $75\pm86$ m$^{-2}$ in the three eutrophic lakes (Fig. 1, Table S3). Other macrofauna, e.g. copepods, *Daphnia*, and leeches
were only occasionally found, and will not be discussed further.

The depth distributions of oligochaetes and chironomid larvae follow different trends (Fig. 2). Chironomid larvae are most abundant in surface sediment (0-5 cmblf), while oligochaetes occur at high abundances over a greater depth interval (Fig. 2). In Lake Greifen and Lake Baldegg, oligochaetes are present at high numbers to 12 and 15 cm sediment depth, respectively, including layers that are distinctly laminated (see horizontal lines in Fig. 2). In Lake Zug, oligochaetes are present to even
greater depths (22 cm). In sediments of Lake Zurich, where oligochaetes and chironomids occur at similar abundances, chironomids dominate the top ~2-3 cm, whereas oligochaetes dominate below. Light microscopic images of the two dominant macrofaunal groups and depth distributions of individual macrofaunal species can be found in the SI (Fig. S1; Table S5).

### 3.2 Macrofaunal community structure and diversity across lakes

Oligochaetes and chironomid larvae were assigned to 9 and 14 different taxonomic groups, respectively (Fig. 3; for station-
specific data, see Fig. S2). All oligochaetes belong to the family *Naididae* (Syn. *Tubificidae*) and all chironomid larvae to the family *Chironomidae*. Two oligochaete morphotypes, *Tubificidae* +bristles and *Tubificidae* -bristles, could not be assigned to a known genus.

Oligochaete group overlap between lakes. Four of the nine groups (*Tubificidae* +bristles, *Tubificidae* -bristles, *P. hammoniensis*, *L. hoffmeisteri*) occur in 4/5 lakes. *E. velutinus* (Lake Zurich), *L. profundicula* (Lake Baldegg) and *P. vejdovskyi*
(Lake Lucerne) were the only species that were only found in one lake. Comparing the dominant oligochaete groups reveals dominance of *Tubificidae* (+bristles) in Lake Zurich, Lake Zug, and Lake Greifen, but very different communities in Lake Baldegg, whichh is dominated by *Tubificidae* (-bristles) and *L. hoffmeisteri*. While *Tubificidae* (+bristles) and *Tubificidae* (-bristles) are uncharacterized, *L. hoffmeisteri* is known to be common in eutrophic to hypereutrophic lakes and highly tolerant





of low-$O_2$ conditions (Brinkhurst, 1982). All identified tubificids except *E. velutinus* are subsurface deposit feeders that are believed to mainly feed on sedimentary bacteria, whereas *E. velutinus* is a surface deposit feeder (Table S4).

Chironomid larval communities in Lake Zurich and Lake Lucerne share many of the same members, but the dominant groups only partially overlap. Lake Zurich sediment is dominated by *Micropsectra sp.*, *Tanytarsus sp.*, *Chironomus*
*riparius/piger gr.,* and *S. coracina,* whereas Lake Lucerne is dominated by *Procladius* sp., *Micropsectra* sp., *M. fehlmanni*, *Tanytarsus* sp., and *S. coracina*. *Micropsectra sp.*, *Tanytarsus sp., and S. coracina* are mainly sedimentary detritus feeders, whereas *Chironomus riparius/piger gr.* is known to mainly filter feed. Both *Procladius* sp. and *M. fehlmanni* are predators (Table S4).

**3.3 C isotope composition of macrofauna and bulk C pools**

Average C isotope compositions of macrofaunal specimens are displayed with those of the potential C sources methane, TOC and DOC in Fig. 4 (for depth profiles, see Fig. S3). Macrofaunal values are lowest in Lake Baldegg (oligochaetes: -36.7±3.3‰, N=14; larvae: -37.6±1.9‰, N=4) and Lake Greifen (oligochaetes: -37.6±2.5‰, N=12; no larvae found) and highest in Lake Lucerne (oligochaetes: -31.7±0.4‰, N=2; larvae: -31.5±2.2‰, N=24) and Lake Zurich (oligochaetes: -32.8±0.9‰ N=5;
larvae: -32.5±2.1‰, N=24). There was no apparent trend between $\delta^{13}$C-values of macrofauna and sediment depth (Fig. S3).

Average $\delta^{13}$C-methane values are in all cases ~35 to 50‰ more negative than those of macrofauna. The most negative methane values are present in Lake Lucerne (-78.8±4.3‰, N=18) and Lake Zurich (-76.7±2.4‰, N=25) followed by Lake Baldegg (-74.3±2.6‰, N=20), Lake Greifen (-73.6±3.7‰, N=21) and Lake Zug (-70.1±4.5‰, N=23). All stations except the middle station in Lake Baldegg have $^{13}$C-methane increases indicative of methane oxidation in surface layers (Fig. S3).
The $\delta^{13}$C-values of TOC are much closer to those of macrofauna (Fig. 4; Fig. S3), with averages ranging from equal (Lake Zurich) to ~5‰ higher (Lake Baldegg). The lowest average $\delta^{13}$C-TOC was measured in Lake Greifen (-34.5±1.5‰, N=35) followed by Lake Baldegg (-32.4±1.2‰, N=37), Lake Zurich (-32.2±1.9‰, N=29), Lake Zug (-30.8±1.3‰, N=35), and Lake Lucerne (-29.7±1.2‰, N=32). Isotopic values of TOC increase by 4-6 ‰ with sediment depth at all sites (Fig. S3). Despite the small differences between $\delta^{13}$C-TOC and $\delta^{13}$C-macrofauna, $\delta^{13}$C-TOC values are significantly higher than those
of oligochaetes and larvae in all lakes except Lake Zurich (Fig. 4).     Average $\delta^{13}$C-DOC is slightly higher than $\delta^{13}$C-TOC in all lakes, and significantly higher than the $\delta^{13}$C of macrofaunal biomass (Fig. 4). Additional analyses on water column algal material and algae bloom layers in sediment (Fig. S3 and Table S2) suggest $\delta^{13}$C-values similar to those of TOC.

**3.4 Average contributions of methane-derived carbon and TOC to macrofaunal biomass C**

A two end member mixing model suggests that on average ≥88% of macrofaunal biomass-C can be explained with assimilation
of detrital organic C (TOC) (Table 3). By contrast, methane-derived carbon accounts for ≤12.1% or ≤6.3% of biomass-C depending on the assumed isotopic fractionation factor during aerobic methane oxidation (for further details see Table 3 caption). Chironomid larvae and oligochaetes from the same lakes have highly similar average methane-derived carbon contributions to biomass. Consistent with past studies (Hershey et al., 2006; Jones and Grey, 2011), the contribution of methane-derived carbon to macrofaunal biomass increases with trophic state, with lowest contributions in Lake Zurich and
Lake Lucerne and highest contributions in Lake Baldegg followed by Lake Zug and Lake Greifen.

**3.5 Microbial communities of macrofauna, larval tubes, and surrounding sediments**

To further investigate the nature of macrofauna-microbiota associations, we studied 16S rRNA gene sequences of macrofauna (whole organisms, guts, residual body without guts) and chironomid larval tubes, and compared these to those in surrounding sediments (Fig. 5).



### 3.5.1 Bacteria

Sediment and tube samples share similar bacterial communites across all lakes, stations, and sediment depths (Fig. 5). Across all locations, sediment and chironomid larval tube samples are dominated by *β-*, *δ-* and *γ-Proteobacteria*, *Chloroflexi* (mainly *Anaerolineae*), *Acidobacteria*, *Bacteroidetes* (dominated by *Sphingobacteriia*), *Planctomycetes*, and *Verrucomicrobia*.

Furthermore, sediments and larval tubes from Lake Zurich and Lake Lucerne share high fractions of *Nitrospirae*. Conspicuous differences are the higher fractions of *δ-Proteobacteria* in sediments and of *Chloroflexi*, *Actinobacteria*, *Gemmatimonadetes*, and *Ignavibacteriae* in tubes, and the virtual absence of *Aminicenantes* in tubes. By comparison, chironomid larvae and oligochaetes have very different bacterial communities, which moreover vary greatly between and within both macrofaunal groups.

Depending on the specimens, bacterial communities of chironomid larvae are dominated by *γ-*, *β-*, and *α-Proteobacteria*, *Firmicutes*, *Actinobacteria, Bacteroidetes*, and/or *Fusobacteria*. Many larval specimens are dominated (>50% of reads) by a single group of *α-*, *β-*, or *γ-Proteobacteria* or *Firmicutes*, and guts of two specimens from Lake Lucerne contain ≥99% *γ-P*roteobacteria. There is no clear trend in relation to lake, trophic state, or water depth. Yet, gut, and to a lesser extent body, bacterial communities from the same samples are sometimes highly similar. Furthermore, bacterial communities in guts

differ clearly from those in the remaining body. For instance, *Firmicutes* in several specimens dominate larval guts, but are virtually absent from the rest of the body. By contrast, the fractions of *α-* and *β-Proteobacteria* are often lower in guts than the remaining body. Compared to larval tubes, chironomid larvae generally have lower abundances of *Chloroflexi* (nearly absent), *Verrucomicrobia*, *Gemmatimonadetes*, *Nitrospirae*, and/or *Ignavibacterieae*.

Bacterial communities of oligochaetes are also variable and differ clearly from those in chironomid larvae, larval

tubes, or sediment. As for chironomid larvae, these bacterial communities do not follow clear trends related to lake, trophic state or water depth. About half of all specimens are strongly dominated (≥80% of 16S reads) by *Fusobacteria* (*Fusobacteriales*), a phylum that accounts for <1% of reads in all tube or sediment samples and was only detected in ~20% of larval specimens. Several other oligochaete specimens are dominated (>50%) by single groups of *α-*, *β-*, *δ-*, and *ε-Proteobacteria*, or *Parcubacteria*, or have high abundances of *Spirochaetae*, and one specimen from Lake Zug is dominated

by *Cyanobacteria*. With the exception of Bacteroidetes, many phyla that are abundant in sediment and/or larval tubes (*Chloroflexi*, *Acidobacteria*, *Gemmatimonadetes*, *Nitrospirae*, *Verrucomicrobiae*, *Aminicenantes*) are less common or nearly absent from oligochaetes. Unlike chironomid larvae, no systematic phylogenetic differences between guts and the rest of the body were detected in oligochaetes. This may, however, be due to the greater difficulty of separating guts from the rest of the body in oligochaetes compared to chironomids.

Ordination plots based on PCoA at the order level (Fig. 6) and at the phylum, class, family and genus level (Fig. S5) confirm the trends observed in Fig. 5. Sediment and tube samples from all lakes and sediment depths are highly similar and form tight clusters, which only become separated at the order level and below. Chironomid larvae and oligochaetes are phylogenetically very different from sediments and tubes, and phylogenetically highly heterogeneous due to dominance by Fusobacteria or *α-Proteobacteria,* or varying relative abundances of diverse proteobacterial classes and orders.

### 3.5.2 Archaea

Archaea only account for low percentages (<10%) of prokaryotic 16S rRNA gene sequences in chironomid larvae, larval tubes, and oligochaetes and were even below detection in 69% of chironomid larval and 39% of oligochaete samples analysed (Fig. S4; also see following section). Yet, distinct trends are evident. Larval tubes have a lower diversity than sediments, being dominated by *Woese-, Pace-* and *Thaumarchaeota* and to a lesser degree *Diapherotrites*. In sediments, *Eury-* and

*Bathyarchaeota* were additionally present at high relative abundances along with lower relative abundances of *Altiarchaeales*, *Lokiarchaeota* and an unclassified phylum-level cluster of *Asgardarchaeota*. In contrast, the archaeal community of larvae was highly variable and dominated by *Pace-*, *Eury-* and *Woesearchaeota*, with typically only 1-2 phyla present per sample.



The oligochaete archaeal community was more diverse and dominated by essentially the same groups as sediments, i.e. *Woese-*, *Pace-*, *Bathy-*, *Eury-* and/or *Thaumarchaeota*, and to a lesser degree *Lokiarchaeota*, *Altiarchaeales*, and *Diapherotrites*.

**3.6 Abundance analysis of Bacteria, Archaea and functional genes related to methane-cycling**

We compared the contributions of Bacteria, methane-cycling Archaea, and methane-oxidizing bacteria across sample types based on ratios of bacterial 16S rRNA gene, *mcrA*, and *pmoA* to total 16S rRNA gene copies. Trends related to lake trophic state are largely absent, but we observe other trends.

Bacteria account for >80% of total 16S gene copies in all samples (Fig. 7, left panel), however, significantly higher proportions are present in oligochaetes, larvae and tubes relative to sediments (Table 4). The contribution of Bacteria decreases
from 94-98% in surface sediments to 82-86% below 12 cmblf. By comparison, Bacteria contribute ≥99% in most macrofauna samples. The lowest bacterial contributions are ~98% in chironomid larvae, 90% in oligochaetes, and 96% in tubes.

In the vast majority of samples, *mcrA* are ≥100 times lower than total 16S rRNA gene copy numbers (range: below the detection limit of ~0.0001% to 2%) (Fig. 7, mid panel). Furthermore, *mcrA* contributions are significantly higher in sediments compared to oligochaetes, larvae and tube samples (Table 4). *mcrA* was even below qPCR detection in chironomid
larvae except one gut sample. While the contribution of *mcrA* increases with depth in larval tubes, oligochaetes and sediments show no depth-related trends. 16S rRNA genes of methane-cycling Archaea were found in sediments (mainly *Methanobacteria* and *M. fastidiosa*), and at very low read numbers in a few tubes (*M. fastidiosa*) and oligochaetes (*M. fastidiosa*, *M. peredens*), but not in larvae.

*pmoA* contributions range from below detection (≤~0.001%) to ~15% (Fig. 7, right panel), and are significantly higher
in oligochaetes but not in larval specimens or larval tubes compared to sediments (Table 4). As for *mcrA*, *pmoA* was only detected in very few (2) larval samples. While *pmoA* contributions decrease with depth in sediments, there is no clear depth trend in oligochaete or larval tube samples. 16S rRNA gene sequences indicate that all methane-oxidizing bacteria are *Gammaproteobacteria*, and are dominated by *Crenothrix* (*Methylococcales*), which are furthermore the only methane-oxidizing bacteria detected in oligochaetes. In addition, low read percentages of *Methylobacter*, *-caldum*, *-coccus*, and *-*
*paracoccus* (all *Methylococcaceae*) occur in larvae, larval tubes and sediments. Additionally, the denitrifying methanotroph *Methylomirabilis* (Candidate phylum NC10) was detected in several tube and sediment samples with highest read percentages in Lake Lucerne.

## 4 Discussion

Methane has been indicated as an important C source to sedimentary macrofauna via grazing on methane-oxidizing bacteria
(Kankaala et al., 2006; Deines et al., 2007a; Jones et al., 2008; Jones and Grey, 2011). Yet, questions remain regarding the conditions under which methane-derived carbon becomes an important C source and how methane-derived carbon is incorporated into macrofaunal biomass. We investigate these questions based on analyses of macrofaunal community structure, isotopic compositions of macrofauna and possible C sources, and microbial community structure across five temperate lakes that range in trophic state from oligotrophic to highly eutrophic.

We observe a clear macrofaunal community shift across the five lakes, with oligochaetes dominating the eutrophic lakes, chironomid larvae dominating the oligotrophic lake, and similar abundances of both macrofaunal groups in the mesotrophic lake (Fig. 1). Maximum abundances of oligochaetes are higher than those of chironomid larvae, and oligochaetes are on average present in deeper sediment layers than chironomid larvae, consistent with different feeding behaviours of the two groups (Fig. 2). Taxonomic analyses reveal overlaps but also clear differences in oligochaete and chironomid larval
communities between lakes (Fig. 3). While oligochaete communities vary strongly with water depth in the same lakes,



oligochaete communities are more uniform across different locations within the same lake. This suggests that chironomid larval and oligochaete communities are controlled by different environmental variables.

Comparing average macrofaunal C isotopic compositions to possible C sources such as methane and TOC, [13]C-methane is always far more negative (-35 to -50‰) while [13]C-TOC is similar or slightly enriched (+0.3 to +5.2‰) relative to

macrofauna. This suggests that organic matter is the main C source of macrofauna (Fig. 4). Estimated contributions of methane-derived carbon range from statistically insignificant to at most 12% and increase with lake trophic state (Table 3). Despite differences in feeding behaviour and environmental drivers behind their species compositions, the contribution of methane-derived carbon is highly similar across chironomid larval and oligochaete specimens from the same lakes, suggesting an important role of lake-specific variables.

Bacterial communities of macrofauna are highly variable, but differ clearly from those in chironomid tubes or sediments. The majority of reads in many macrofaunal specimens belong to single ZOTUs, suggesting potential symbiotic interactions with their hosts (Fig. 5, Fig. 6; discussed in detail later). Consistent with C isotope data, *pmoA* copy numbers indicate that these bacterial symbionts are largely not methane-oxidizing bacteria. Yet, *pmoA* contributions in several oligochaetes, chironomid larvae and larval tubes are elevated compared to surrounding sediment (Fig. 7, right panel; Table 4),

consistent with methane-oxidizing bacteria contributing a minor fraction of biomass C. By contrast, grazing on methanogenic or anaerobic methane-oxidizing archaea does not explain the observed isotopic values of larvae and oligochaetes. *McrA* contributions were always small (≤1%) and significantly lower in oligochaetes, chironomid larvae, and chironomid larval tubes than in surrounding sediment (Fig. 7, mid panel; Table 4).

In the following sections, we discuss in detail the potential drivers of macrofaunal community structure, the trophic

role of macrofauna, and the potential trophic roles of observed (endo)symbiotic bacteria in their macrofaunal hosts.

## 4.1 Abundance and taxonomy of macrofauna along trophic state

Oligochaete abundances follow the environmental index proposed previously by Milbrink (1983), which predicts a strong rise in worm abundance with increasing trophic state. Chironomid abundances are also within the range previously reported for lakes (Mousavi, 2002). While chironomid larvae show typical depth distributions (e.g. Panis et al., 1996), oligochaetes have

unusually deep ranges. While most studies report that oligochaetes are mainly present at 2-8 cm sediment depth (reviewed in McCall and Tevesz, 1982), we observe high worm abundances to 10-14 cm in the eutrophic lakes.

The shift in dominance from chironomid larvae to tubificid oligochaetes with increasing trophic state (Figs. 1 & 2) is in line with previous studies. Oligochaetes frequently dominate eutrophic lakes (Saether, 1980; Lang, 1985; Timm, 1996; Bürgi and Stadelmann, 2002), and changes from chironomid larvae- to oligochaete-dominated communities have been reported

as one of the first signs of eutrophication (Saether, 1979). The observed dominance of oligochaetes in eutrophic lakes is possibly related to an overall higher tolerance of low $O_2$ conditions, as many oligochaetes feed in anoxic parts of sediments (McCall and Tevesz, 1982) and efficiently exchange gases through their body walls (Martin et al., 2008). An on average longer survivorship of anoxic conditions has also been reported for oligochaetes compared to chironomid larval species (Hamburger et al., 1998). Nonetheless, certain species of chironomid larvae can respire anaerobically and tolerate weeks of anoxia (Pinder,

1995). Thus, the dominance of oligochaetes in eutrophic lakes could also be due to factors other than $O_2$ concentrations, e.g. superior ability to exploit high organic matter deposition rates, better protection from benthic predators, such as bottom-feeding fish, which are abundant in eutrophic lakes (Scheffer et al., 1993), due to deeper burrows, or interspecific interactions between oligochaetes and chironomid larvae.

The distributions of taxonomically classifiable oligochaete species match those in previous studies. *L. hoffmeisteri*, a

species that is widespread and common in eu- to hypertrophic lakes, which feeds on sedimentary bacteria and algae and is very tolerant of low-$O_2$ concentrations (Brinkhurst, 1982), occurs at high abundances in Lake Baldegg (Table S5). *P. hammoniensis* and *T. tubifex*, which frequently co-occur at high abundances in meso- to eutrophic lakes (Lang, 1990; Timm, 1996), were





dominant groups in Lake Zurich, Lake Zug, and/or Lake Greifen. On the other hand, *E. velutinus*, a surface-deposit feeder that indicates oligo- to mesotrophic conditions (Martin et al., 2008), was only found in Lake Zurich. Notably, however, most oligochaete specimens found in this study could only be classified to the family-level based on the presence/absence of bristles (*Tubificidae* (+bristles); *Tubificidae* (-bristles); Fig. 3). Detailed insights into the behaviours of these groups are not possible

except that both occurred over the entire depth interval where worms were found (Table S5).

Remarkably, the lakes investigated show little evidence of sediment mixing even though tubificids are known to be subsurface conveyor feeders that cause strong sediment mixing (Fisher et al., 1980; Matisoff et al., 1999). We observed clear laminations at the deep station in Lake Baldegg and the deep and middle station in Lake Greifen in sediments that were deposited in the mid 1980s and ~2010, respectively (Figs. 2 & S7; Fiskal et al., 2019). The distribution of laminated layers at

these stations thus matches the years when artificial water column mixing and oxygenation were initiated in these lakes (Lake Baldegg: 1984; Lake Greifen: 2009; Fiskal et al., 2019), and suggests rapid re-colonization by macrofauna after the onset of aeration. Yet, based on the continued presence of lamina in sediments deposited until the onset of aeration, it appears that mixing has been exclusive to surface sediments. Depth profiles of radionuclides confirm this interpretation and even indicate minimal sediment mixing across all stations through time (Fig. S7). At all stations, $^{137}$Cs peaks that match the 1986 (Chernobyl)

and 1963 (bomb test) time markers, are present and $^{210}$Pb$_{unsupported}$ clearly decreases from the top 2 cm downward. These findings contrast with previous studies which showed rapid homogenization of sediment to 10 cm by tubificids in the laboratory (Fisher et al., 1980; Matisoff et al., 1999) and homogeneous radionuclide profiles to 6 cm in tubificid-dominated natural lake sediments (Robbins et al., 1977; Krezoski et al., 1978).

The community composition of chironomid larvae also shows differences in relation to trophic state (Fig. 3). Large,

free-living and predatory chironomid larvae account for approximately half of the specimens from Lake Lucerne, whereas tube-building herbivorous, surface detritus-feeding and/or gardening larvae dominate Lake Zurich and the small sample sizes in eutrophic lakes (Fig. 3; Table S5). The shift in diet at higher trophic levels matches the higher sedimentary input of algae and algal detritus under these conditions (Fiskal et al., 2019), whereas the potential increase in microbial gardening matches switches in feeding behavior by *C. riparius* and other *Chironomus* spp. under hypoxic or eutrophic conditions (Stief et al.,

2005; Yasuno et al., 2013). By contrast, the reasons for the increased relative abundances of predatory chironomid larvae, in particular at the shallow and mid station in Lake Lucerne (Fig. S2), are unknown. Plausible reasons are lower hypoxia tolerance of large predatory *Macropelopia* and *Procladius* spp. (Hamburger et al., 1998; Brodersen et al., 2008), and higher availability of zooplankton food due to reduced planktivory by fish in oligotrophic lakes (Jeppesen et al., 1990; Jeppesen et al., 1999). In addition, large free-living chironomids may especially suffer from predation in meso- and eutrophic lakes, where populations

of bottom-feeding fish that feed on chironomid larvae frequently increase due to reduced piscivory by larger fish (Scheffer et al., 1993).

### 4.2 Carbon sources of lake sedimentary macrofauna

Similar to previous studies (e.g. Grey et al., 2004; Jones et al., 2008) we observe an apparent increase in the contribution of methane-derived carbon with increasing trophic state. Yet, we calculate this contribution to be at most 12%, even in the highly

eutrophic Lake Greifen and Lake Baldegg. By comparison, other studies have estimated methane-derived carbon contributions of >40% in chironomid larvae in eutrophic lakes (e.g. Deines and Grey, 2006; Eller et al., 2007; Jones et al., 2008) and reported strong δ$^{13}$C-depletions in oligochaete specimens from profundal sediment (Premke et al., 2010). Nonetheless, minor contributions of methane-derived carbon to the biomass of benthic invertebrates are not new. A survey of 87 lakes suggested that marked $^{13}$C-depletions were only present in chironomid larvae from lakes with seasonal stratification and bottom water

anoxia (Jones et al., 2008). Moreover, the limited published δ$^{13}$C data on lake oligochaetes are mostly similar to those of TOC (Kiyashko et al., 2001; Premke et al., 2010).





In support of C-isotopic interpretations, DNA-based analyses indicate that neither methane-oxidizing bacteria nor methanogens are dominant microorganisms in surface sediments or chironomid larval tubes. Thus, strong enrichment or gardening of methane-oxidizing bacteria or methanogens as observed elsewhere in chironomid tubes (e.g. Kajan and Frenzel, 1999; Kelly et al., 2004) or surface sediments (e.g. Eller et al., 2005; Deines et al., 2007a) is absent. Yet, the reasons are

unclear. Despite being artificially oxygenated, bottom water in Lake Baldegg and Lake Greifen experiences seasonally low $O_2$ conditions (0.5-4 mg L$^{-1}$) or hypoxic conditions (<0.5 mg L$^{-1}$), respectively (Fiskal et al., 2019). These values are within or below the seasonal $O_2$ threshold (2-4 mg L$^{-1}$) that is characteristic of lakes with marked $^{13}$C-depletions in chironomid biomass (Jones et al., 2008). Jones et al. (2008) argued that the contribution of methane-derived carbon increases inversely with the depth of the oxic-anoxic interface. In June 2016, this interface was only ≤1 mm at all stations in Lake Baldegg and ≤2 mm in

Lake Greifen, while the zone of methanogenesis extended into the top 1 cm of sediment (Fiskal et al., 2019). Thus, conditions were potentially well-suited for a high contribution of methane-derived carbon. It is possible that methane-derived carbon mainly plays a role in ventilating and tube-building chironomid larvae because stable and narrow oxic-anoxic interfaces (Brune et al., 2000), combined with high $O_2$ and methane supplied by ventilation, create excellent growth conditions for methane-oxidizing bacteria . By comparison, tubificids do not produce stable structures that promote the establishment of steady oxic-

anoxic interfaces and perform less burrow ventilation than chironomid larvae (Gautreau et al., 2020 and references within). Yet, the fact that all three identified larvae from Lake Baldegg belong to tube-building taxa, and that the four isotopically analysed larval specimens from this lake only had minor methane-derived carbon contributions suggests that additional, still unknown factors control methane-oxidizing bacteria enrichment by tube-building chironomids in surface sediment.

Our C-isotopic data indicate that algal or detrital organic carbon, or microorganisms that feed on and acquire the

isotopic signatures of algal or detrital organic carbon, are the main food sources of dominant macrofauna. Selective feeding on "isotopically light" subportions of the TOC pool, rather than methane-derived carbon, could even explain the observed slight isotopic depletions of oligochaetes and chironomid larvae in eutrophic lakes. Under this scenario, isotopic data on phytoplankton-dominated sediments might provide useful indications. Yet, our limited data on algal bloom layers in sediments and phytoplankton from overlying water indicate similar $^{13}$C-values relative to TOC (Fig. S3). Other potential explanations,

e.g. preferential feeding on organic C from surface sediments, which in many cases have the most negative C-isotopic values, or isotopic fractionations during C-assimilation and biosynthesis are also unlikely. Oligochaetes are widely reported to feed at several centimeters depth within the reduced part of sediments (McCall and Tevesz, 1982), and C-isotopic fractionation during biosynthesis of bulk animal biomass are typically low (Fry and Sherr, 1989).

### 4.3  Potential diet and host-microorganism interactions in tubificid worms

Shallow subsurface, head-down deposit-feeding on detrital particles by tubificids typically leads to significant mixing of surface sediments (Fisher et al., 1980; Rodriguez et al., 2001). Yet, since sediment reworking is minimal in the lakes studied, deposit-feeding may not be the main dietary mode. Instead, the unusually deep distributions may provide clues, e.g. oligochaetes could selectively graze on microbial biofilms inhabiting the walls of their deep and extensive gallery-type burrow networks. Under this scenario one might expect to find large amounts of DNA of sediment microorganisms within

oligochaetes. This is not the case, however, suggesting that grazed microbial communities are either very different from those in sediments or that the DNA of ingested microorganisms is rapidly degraded, and thus overshadowed by DNA pools of endosymbionts.

Another foraging strategy may not involve ingestion through the oral cavity. *T. tubifex* actively take up short-chain organic acids, such as acetate and propionate, from their surroundings through their body wall (Hipp et al., 1985; Sedlmeier

and Hoffmann, 1989). The subsequent respiration of these organic acids can account for up to 40% of *T. tubifex* energy turnover (Hipp et al., 1986). In other species of tubificids, uptake of amino acids through the body wall has been reported (Brinkhurst and Chua, 1969). Tubificid body walls are, moreover, permeable to dissolved gasses, which is why oligochaetes acquire much



of their $O_2$ by undulating movements of their tail ends in oxic water above sediments (Brinkhurst, 1996). This permeability to gases could, in principle, also provide energy, e.g. if gases, such as methane or $H_2$, diffusing from sediment pore water into oligochaetes are taken up by endosymbiotic microorganisms. Thus, the unusually deep distributions in the lakes studied could indicate that oligochaetes are using the products of microbial hydrolysis (amino acids), fermentation (short-chain organic acids,

$H_2$), and respiration (methane) as energy sources.

Matching the slight increase in methane-derived carbon in eutrophic lakes, we observe significantly higher contributions of *pmoA* in oligochaetes compared to surrounding sediment. Assuming that these *pmoA* belong to living methane-oxidizing bacteria, the presumed shuttling of oligochaetes between methane-rich deep sediment layers and the oxic sediment surface may result in favourable conditions for methane-oxidizing bacteria and produce a symbiotic relationship that could be

called "internal microbial gardening". The potential of annelid hindguts to make excellent microbial habitats was demonstrated in the polychaete *Abarenicola vagabunda* (Plante et al., 1989). How this methane-derived carbon would be assimilated by oligochaetes is unclear, however. Potential mechanisms include uptake of organic intermediates of methane oxidation, e.g. methanol, through the hindgut and ingestion of oligochaete faeces that are enriched in methane-oxidizing bacteria.

Given the nonetheless minor contributions of methane-oxidizing bacteria to the microbiota of oligochaetes, and the,

on the other hand, high contributions of single, non-methane-oxidizing bacteria ZOTUs to the microbial communities of these organisms, questions arise concerning the roles of these dominant ZOTUs. We observe that in 22 of the 30 oligochaete specimens sequenced a single ZOTU accounted for >50% of the total reads (Table S6). In 15 specimens, these ZOTUs belong to a single genus-level cluster of unclassified *Fusobacteriaceae* (Fusobacteriaceae Cluster I) that was previously found in earthworm and aquatic vertebrate intestines, anaerobic sediments, bioreactors, soil and diverse water samples (Fig. S8; Table

S6). The high percentages of this cluster are especially striking considering that *Fusobacteriaceae* account for on average only 0.01±0.02% of total 16S reads in sediments. All cultivated members of Fusobacteriaceae are anaerobes that fermentatively degrade polymeric organic compounds, in particular proteins and carbohydrates, with acetate, butyrate, and other short-chain organic acids as main end products (Olsen, 2014). Given previous evidence for the preference of proteinaceous organic matter by tubificids (de Valk et al., 2017), these *Fusobacteriaceae* could be primary degraders of proteins within the digestive tract

of oligochaetes. This relationship could be mutually beneficial, commensal, or parasitic. Yet, given their widespread dominance, a mutualistic relationship, where the host respires organic acids produced by the symbiont, and the symbiont benefits from the input of organic substrates and maintenance of low end product concentrations by the host, seems likely.

The remaining six dominant ZOTUs in oligochaetes fall into the phyla *Proteobacteria (α, β, and ε-classes)*, *Bacteroidetes*, and *Parcubacteria* (Fig. S8, Table S6). ZOTU18 falls into the anaerobic *ε*-proteobacterial genus *Wolinella*

(order *Campylobacterales*), isolates of which use $H_2$ or formate as electron donors and fumarate and nitrate as electron acceptors (Tanner and Paster, 1992). The production of succinate, which is the main end product of fumarate reduction by *Wolinella*, could benefit hosts under low $O_2$ conditions, given that succinate is the main energetic intermediate during anaerobic metabolism of tubificids (Seuβ et al., 1983). ZOTU8 falls into the facultatively aerobic *β*-proteobacterial genus *Deefgea* (order *Neisseriales*), members of which ferment carbohydrates to organic acids (Stackebrandt et al., 2007). This ZOTU may provide

energy to its host in the same way proposed for *Fusobacteriaceae*. The *α*-proteobacterial ZOTU4 falls into the family *Holosporaceae*, members of which are obligately intracellular, potentially parasitic symbionts of ciliates (Santos and Massard, 2014). This ZOTU dominated one oligochaete from each Lake Baldegg and Lake Zug and could derive from commensal ciliates, which frequently inhabit the guts of freshwater oligochaetes (Falls, 1972). Alternatively, given the very high percentage of total 16S reads in one entire worm specimen (93%), a novel form of (intracellular) symbiosis with oligochaetes

cannot be discounted. Similarly unclear is the host relationship with ZOTU199, which belongs to the candidate phylum *Parcubacteria* of the enigmatic Candidate Phyla Radiation (Brown et al., 2015). DNA sequences of this candidate phylum have been retrieved from diverse, mostly anoxic habitats, have genes that are linked to the fermentation of carbohydrates, and have been implicated with ectosymbiotic or parasitic lifestyles (Wrighton et al., 2012; Nelson and Stegen, 2015). The



remaining ZOTUs fall into an unclassified genus-level subcluster of *ß-Proteobacteria* (ZOTU6; order *Rhodocyclales*) and an unclassified order-level cluster of *ɛ-Proteobacteria* (ZOTU9). Based on existing knowledge, it is not possible to infer the potential roles of these ZOTUs within their hosts.

### 4.4 Potential host-microorganism interactions in chironomid larvae

Similar to tubificids, the majority of chironomid larvae (12/19 sequenced specimens) are dominated by single ZOTUs (Table S6). Interestingly, more specimens are dominated by single ZOTUs in Lake Lucerne (9/10) than in Lake Zurich (3/7) or Lake Baldegg (0/2), suggesting that the frequency, and possibly importance, of these associations is inversely related to trophic state. The dominant ZOTUs belonged mainly to the phylum *Proteobacteria* (10/12; *α, ß,* and *γ*-classes). In addition, the same unclassified *Fusobacteriaceae* (Fusobacteriaceae Cluster I) that dominated oligochaetes, and an unclassified sister group of *Bacteroides (Bacteroidetes)*, which we call "Unclassified Wastewater & Gut Group" based on its reported occurrences, dominated each one larval specimen.

Two proteobacterial groups most commonly dominate chironomid larvae. ZOTU2 of the genus *Wolbachia* (*Rickettsiales*) dominates four specimens. Members of this α-proteobacterial genus are widespread intracellular symbionts of insects whose relationships with their hosts range from parasitic to mutualistic (Correa and Ballard, 2016), though dietary contributions have not been demonstrated to our knowledge. ZOTU3 and -21 of the *γ*-proteobacterial genus *Aeromonas* dominate three specimens. Members of this facultative anaerobic genus are widespread in aquatic environments (Huys, 2014) and were previously found in chironomid larvae (Eller et al., 2007) and numerous other aquatic invertebrates (Harris, 1993). Aeromonads can ferment carbohydrates to organic acids, which might supplement the diet of chironomid larvae, but are also animal pathogens, and have been shown to degrade the egg masses of chironomids (Senderovich et al., 2008). Similar functions, ranging from mutualistic to detrimental, are likely for the *γ*-proteobacterial genus *Serratia* (*γ-Proteobacteria*), an unclassified cluster of *γ*-proteobacterial *Pseudomonadales* (family *Moraxellaceae*), and the Unclassified Wastewater & Gut Group. ZOTUs of these groups each dominate one larval specimen (ZOTU11, ZOTU26, and ZOTU28, respectively). All three groups degrade carbohydrates anaerobically (*Serratia*, *Bacteroidetes*) or aerobically/facultatively anaerobically (*Moraxellaceae*) to organic acids, and may thereby provide energy to chironomid larvae, but can also be pathogenic or mutualistic in ways unrelated to diet (Grimont and Grimont, 2006; Sabri et al., 2011; Teixeira and Merquior, 2014; Wexler, 2014). The remaining ZOTUs belong to unclassified genus-level subclusters of *ß*-proteobacterial *Rhodocyclales* (ZOTU6) and *Burkholderiales* (ZOTU12). Due to the very diverse ecophysiologies of members of *Rhodocyclales*, *Burkholderiales*, and *Pseudomonadales*, the potential significance of these ZOTUs to their hosts are highly uncertain.

### 5 Conclusions

Our study indicates clear changes in lacustrine sedimentary macrofaunal communities with increasing trophic state, including a shift in dominance from chironomid larvae to tubificid oligochaetes. By combining C isotope and genetic analyses, we show that, independent of faunal group or trophic state, detritus-derived organic carbon rather than methane-derived carbon is the main cabon source of these animals. Yet, the exact carbon sources remain unclear and may include anything from actual detritus to detrital carbon-assimilating microorganisms or even breakdown products of microbial detritus degradation. Given that two thirds of the symbionts in oligochaetes fall into a family of known protein-degrading bacteria (*Fusobacteriaceae*), selective feeding on protein-rich organic matter fractions, such as microbial cells, is likely for these oligochaete specimens. Similarly, given that half of the dominant symbiont ZOTUs in chironomid larvae belong to carbohydrate degrading taxa (*Aeromonas*, *Serratia*, *Moraxellaceae*, *Bacteroidaceae*), preferential feeding on algae or algal detritus in surface sediments is plausible for these chironomid taxa. Though more research is needed, both macrofaunal groups may benefit from protein- and carbohydrate-degrading endosymbionts through the production of short-chain organic acids, which are taken up through the



hindgut wall and subsequently used for energy conservation by respiration, as well as a precursors for biosynthesis. This benefit is likely greatest if presumed gut symbionts mainly feed on portions of the protein or carbohydrate pools that cannot be degraded by the host.



**Acknowledgments**

We thank the Genetic Diversity Centre (GDC) at ETH Zurich for help with Next Generation Sequencing and analysis. We are thankful to AquaDiptera and AquaLytis for the taxonomic analysis of the macrofauna samples, especially to Susanne Michiels and Ute Michels. We thank Madalina Jaggi for technical support of macrofaunal $^{13}$C analysis and Serge Robert for technical

5     support during $\delta^{13}$C-methane measurements. This project was funded by Swiss National Science Foundation project no. 205321_163371 to Mark A. Lever.

**Data availability**

Biogeochemical data will be made available after publication of the manuscript on Pangaea. The sequence data have been deposited at DDBJ/EMBL/GenBank under the accession KDVU00000000. The version described in this paper is the first

10     version, KDVU01000000.

**Conflict of interest**

The authors declare that they have no conflict of interest.

**Author contributions**

15     AF, AM, EA, LD, XH, RZ, LL, ND, CJS, SMB and MAL helped with sample collection and/or measurements. AF, SMB, and MAL substantially contributed to the interpretation of data. AF and MAL wrote the manuscript. MAL designed the study and acquired the funding for the project. All authors commented on and approved the final version of the manuscript.





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

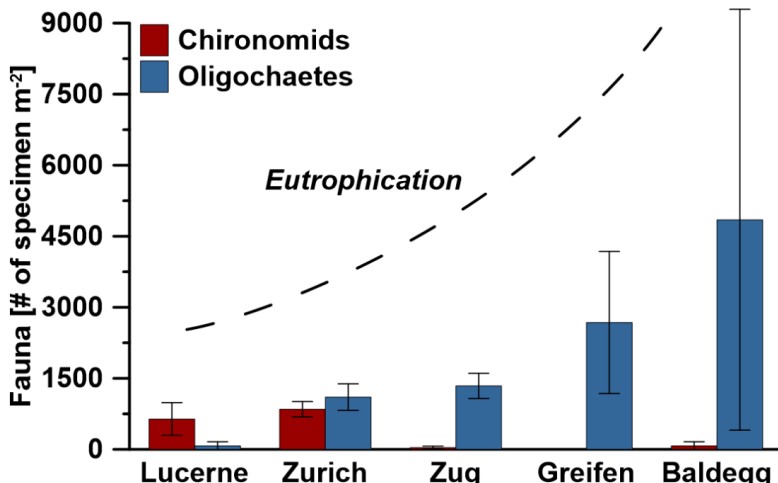

**Fig. 1: Average abundances of macrofauna in each lake. Error bars indicate standard deviations of 3 stations per lake, except for Lake Zurich, where the macrofauna-free deep station was not considered and error bars indicate the range of the two shallower stations.**



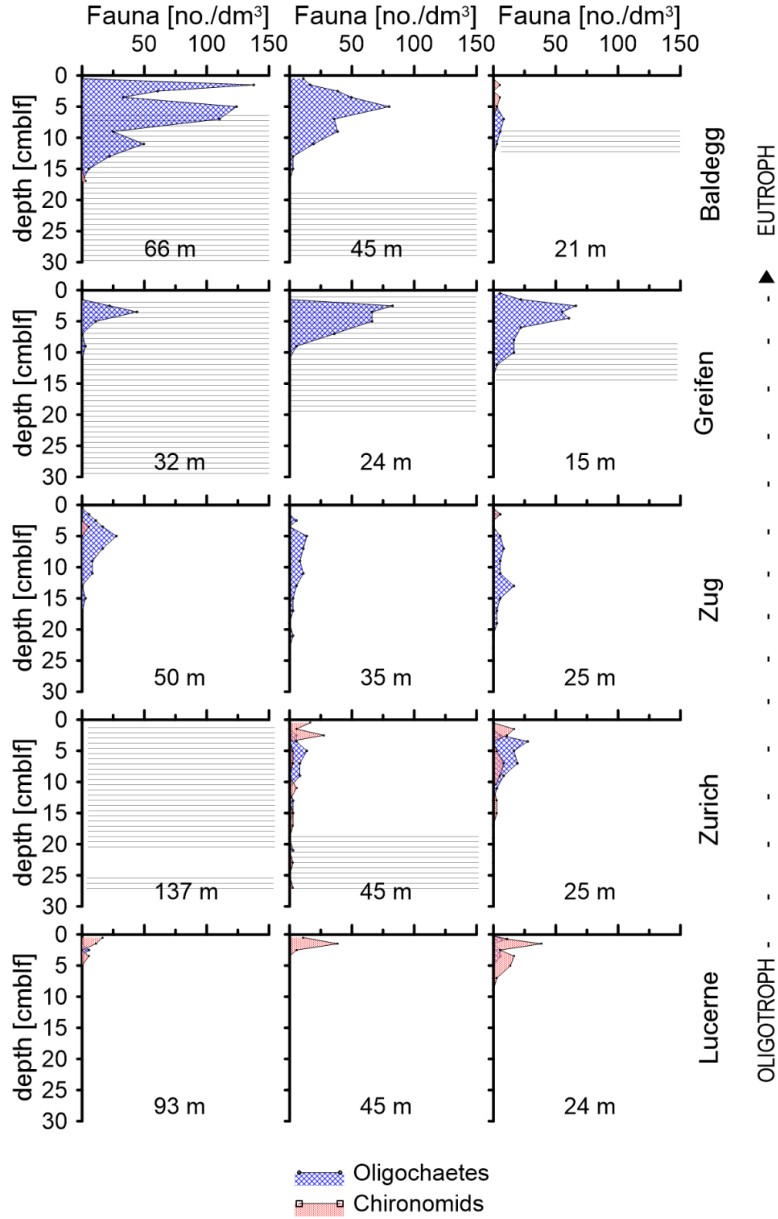

Fig. 2: Depth distributions of oligochaetes and chironomid larvae at each station. Water depths of each station are indicated in each subplot. Horizontal lines indicate depth distributions of laminated sediment layers.



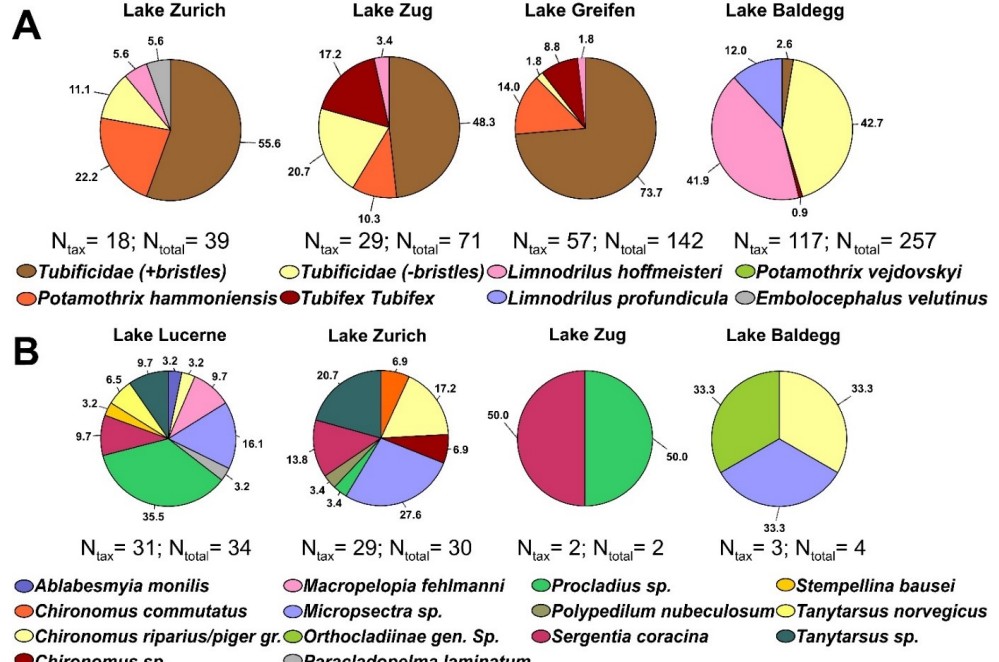

**Fig. 3:** Pie charts of taxonomic analyses on oligochaetes (A) and chironomid larvae (B) in each lake (Ntax=number of taxonomically identified specimens, Ntotal = total number of specimens). No chironomid larvae were found in Lake Greifen. In Lake Lucerne only 4 oligochaetes were found of which 1 was taxonomically analyzed (Potamothrix vejdovskyi; not shown).





**Fig. 4: Boxplots of $^{13}$C isotopic compositions of CH$_4$, TOC, DOC, oligochaetes and chironomid larvae for each lake (note: no larvae were found in Lake Greifen). Boxes show 75% and 25% quartiles. Whiskers show minimum and maximum values. Wilcoxon signed rank tests were applied to check whether $^{13}$C-isotopic signatures of macrofauna and TOC were significantly different (ns=not significant; \*=p<0.05; \*\*= p<0.01; \*\*\*=p<0.001). For each Wilcoxon test, macrofaunal specimens were paired with TOC isotopic signatures from the same depth (±2 cm), and only data were included for which there were data macrofauna and TOC data from matching depths. Samples with N<5 are displayed as individual data points.**

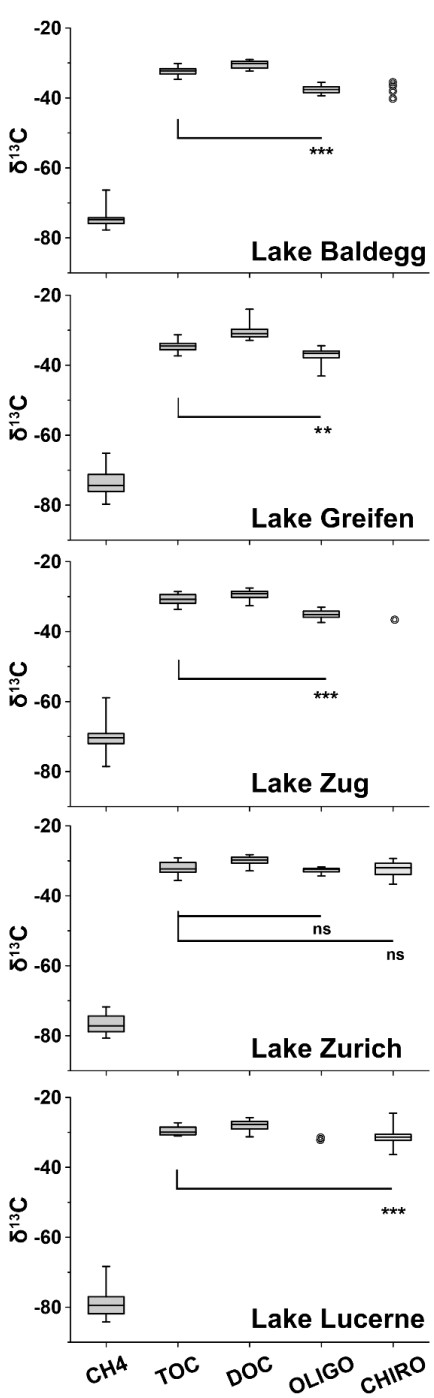







**Fig. 5: Relative abundances of Bacteria at the phylum level (Proteobacteria at class level) based on 16S rRNA gene sequences. Sequences were obtained from 17 sediment, 10 chironomid larval tube, 26 chironomid larvae (Nbody=7, Ngut=7, Nwhole=12), and 36 oligochaete (Nbody=5, Ngut=6, Nwhole=25) samples. Station and sample IDs are indicated by sample names, which indicate station water depth (m), sediment depth (cm) and portion of macrofaunal body analysed (w=whole specimen, g=gut, b=body). Bodies and guts of the same specimens are marked by the same symbols. All sediment 16S rRNA gene sequence data are from Han et al. (2020), in press.**

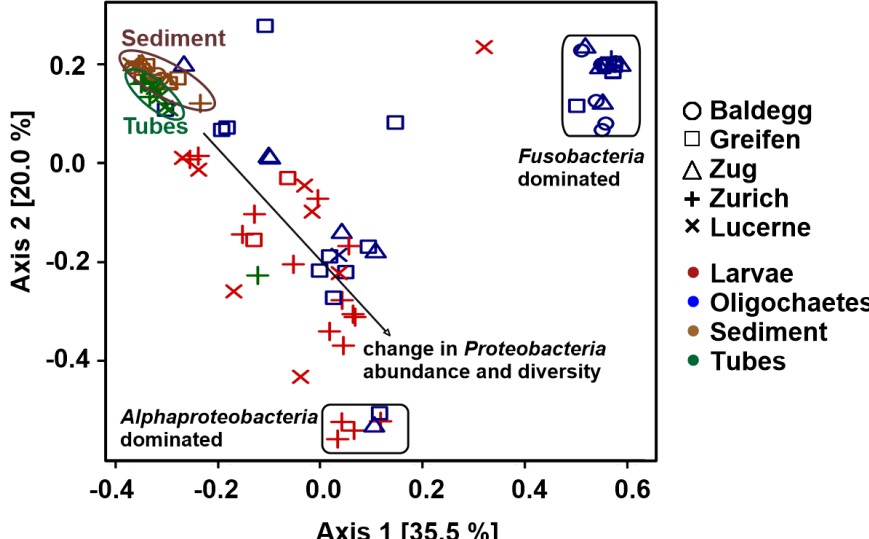

**Fig. 6: PCoA analysis of bacterial community structure at the order level using Bray-Curtis distances.**

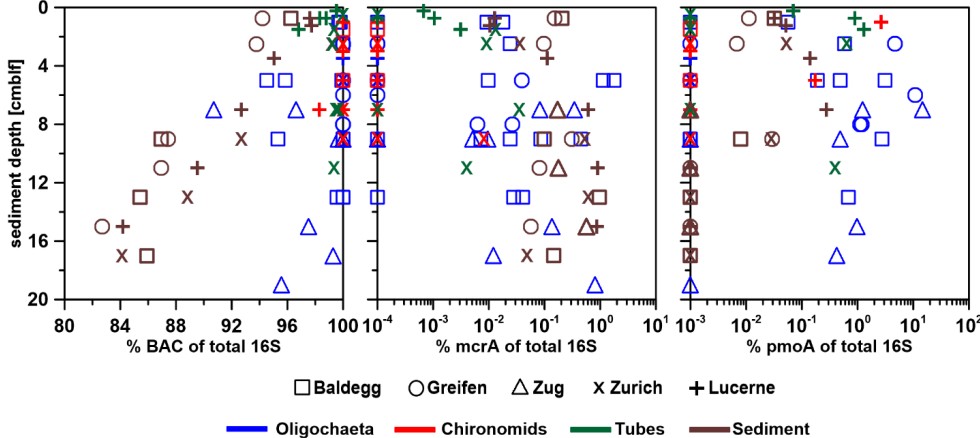

**Fig. 7: Ratios (expressed in %) of bacterial (BAC) 16S rRNA gene copy numbers (left panel), mcrA copy numbers (mid panel), and pmoA copy numbers (right panel) to total 16S rRNA gene copy numbers (sum of bacterial and archaeal 16S rRNA gene copy numbers). The three x-axes differ in ranges and scales (linear and log). All sediment 16S rRNA gene values are from Han et al. (2020). Values on the lower limit of the x-axis in the mid and right panel indicate samples in which mcrA or pmoA were below qPCR detection.**

**Table 1: qPCR primers and standards and their corresponding references**

| Target | Primer | Sequence 5' - 3' | Reference | Standard |
|---|---|---|---|---|
| Archaeal | *Arc915F_mod* | AAT TGG CGG GGG AGC AC | Cadillo-Quiroz et al. (2006) | *Thermoplasma* |
| *16S rRNA* | *Arc1059R* | GCC ATG CAC CWC CTC T | Yu et al. (2005) | *acidophilum* |
| Bacterial | *Bac908F_mod* | AAC TCA AAK GAA TTG ACG GG | Lever et al. (2015) | *Desolfotignum* |
| *16S rRNA* | *Bac1075R* | CAC GAG CTG ACG ACA RCC | Ohkuma and Kudo (1998) | *phosphitoxidans* |
| *pmoA* | *A189F* | GGN GAC TGG GAC TTC TGG | Holmes et al. (1995) | *Methylococcus* |
| | *Mb661R* | CCG GMG CAA CGT CYT TAC C | Costello and Lidstrom (1999) | *capsulatus* |
| *mcrA* | *Mlas_F* | GGT GGT GTM GGD TTC ACM CAR TA | Steinberg and Regan (2009) | *Methanocorpusculum* |
| | *mcrA-rev* | CGT TCA TBG CGT AGT TVG GRT AGT | Steinberg and Regan (2009) | *parvum* |





**Table 2: qPCR protocols for each primer pair.**

| Primer target: | | Arc | Bac | pmoA | mcrA |
|---|---|---|---|---|---|
| qPCR step | time (min:s) | primer-specific temperature (°C) | | | |
| **1. Initial Activation** | 05:00 | Always 95 | | | |
| **2. Denaturation** | 00:10 | Always 95 | | | |
| **3. Annealing** | *00:30* | *55* | *60* | *(62) 52* | *56* |
| **4. Polymerization** | 00:15 | Always 72 | | | |
| **5. Acquisition** | *00:05* | *81* | *82* | *80* | *80* |
| | *Cycle repeats step 2.-5.* | ***40*** | ***45*** | ***(10+) 35*** | ***40*** |
| **6. Denaturation** | 01:15 | *Always 95* | | | |
| **7. Acquisition** | *continuous* | *55-95* | *60-95* | *60-95* | *53-95* |
| **8. Cooling** | ∞ | Always 4 | | | |

**Table 3: Contributions of TOC and methane to oligochaete and chironomid larval biomass C based on a two end member mixing model. Estimates outside of the parentheses are maximum values, as they assume no isotopic fractionation during aerobic methane oxidation. Values within parentheses are more conservative and assume a fractionation factor that is in the upper range previously determined for freshwater sediments and pure-culture incubations (-39‰) (Kruger et al., 2002; Templeton et al., 2006; Kankaala et al., 2007). For the calculations, only macrofaunal specimens were included that could be paired with TOC and methane isotopic values from the same sediment depth (±2 cm).**

| | Contribution of TOC (%) | | Contribution of methane (%) | |
|---|---|---|---|---|
| | Oligochaetes | Chironomid larvae | Oligochaetes | Chironomid larvae |
| **Lake Lucerne** | --- | 97.3±4.1 (98.6±2.0) | --- | 2.7±4.1 (1.5±2.0) |
| **Lake Zurich** | 98.5±3.9 (99.2±1.5) | 99.1±4.3 (99.5±2.4) | 1.5±3.9 (0.8±1.5) | 0.9±4.3 (0.5±2.4) |
| **Lake Zug** | 88.3±3.3 (94.0±1.7) | --- | 11.7±3.3 (6.0±1.7) | --- |
| **Lake Greifen** | 93.1±7.6 (96.5±3.5) | --- | 6.9±7.6 (3.5±3.5) | --- |
| **Lake Baldegg** | 88.2±2.8 (93.9±1.5) | 87.9±1.6 (93.9±0.8) | 11.8±2.8 (6.2±1.5) | 12.1±1.6 (6.3±0.8) |

**Table 4: Results of Wilcoxon sign rank test to examine whether the ratios of bacterial 16S rRNA gene, mcrA, and pmoA to total 16S rRNA gene copy numbers differ significantly between oligochaete, chironomid larval, and chironomid larval tube samples relative to surrounding sediment (ns=not significant; *=p<0.05; **=p<0.01; ***=p<0.001). Only data were included for which matching values existed from the same sediment depth (±2 cm).**

| *%* | Test | Oligochaetes vs. Sediment | Larvae vs. Sediment | Tubes vs. Sediment |
|---|---|---|---|---|
| ***BAC of total 16S*** | greater | *** | *** | ** |
| ***mcrA of total 16S*** | less | ** | *** | ** |
| ***pmoA of total 16S*** | greater | ** | ns | ns |