# Peer review of "Carbon sources of benthic fauna in temperate lakes across multiple trophic states"

_Biogeosciences, 2020_

## Referee Comment (RC1) · Anonymous Referee #1 · 16 Nov 2020

General: In the submitted manuscript by Annika Fiskal et al. the authors sampled sediments from lakes with different levels of eutrophication. The aim was to investigate differences in macrofauna (oligochaetes and chironomids), microbial communities in the sediment as well as on/in the macrofauna, and the contribution of methane derived carbon for macrofauna ingestion/assimilation. The authors found that methane derived carbon is a minor carbon source for macrofauna, and that macrofauna associated prokaryotes are different from sediment prokaryotes.

The conclusions drawn from the stable carbon isotope data is rather uncertain. This data was used to investigate methane derived carbon and if macrofauna ingest or assimilate this in their bodies. The authors do not know the isotope compositional values for potential food sources derived from methane (such as methane-oxidizing

bacteria). From what I have read online methanotroph lipids can range between -45‰ to -65‰ d13C values which is very different from the macrofauna values presented in the submitted manuscript. It would therefore be good if the authors tone down the discussion and conclusions from these findings. The authors can instead focus more on the qPCR data that indicate methane cycling bacteria in the gut of the studied macrofauna. And the isotope data might then be used as supportive data to help support the findings that methane derived carbon is a minor food source.

The discussion is quite long and I think this can be shortened by almost half. The authors go into specific details about the microbiology data on ZOTU level, and paragraphs that mention previous studies with similar results can be shortened. I think the discussion can be better summarized and more focused in relevance to the aim of the study. I also think that the focus on the macrofauna associated bacteria can be shortened in the manuscript, as is the case in the Abstract where it is just mentioned briefly at the end, while a large part of the discussion is dedicated to this subject.

There is also essential information missing in the methods such as DNA extraction from the sediment, bioinformatics, and DNA sequencing. It seems that this part is instead presented in a manuscript that is in press (Han et al 2020), however there is no need to present this data as results for this manuscript. I also think it might be misleading to do so and the authors better double-check the journal guidelines for what is acceptable. Instead the authors can mention relevant findings from Han et al. (2020) in the discussion. If the results are first presented in Han et al. (2020) then it should not be presented again as new results for this manuscript. Furthermore, Han et al. (2020) is missing in the reference list so there is no way for the reviewers to read these methods or results.

I think the authors have a large and interesting dataset and it should definitely be published here or somewhere else. My opinion is that the manuscript needs to be more streamlined and focused on a single story (now it feels like two stories: one geochemical with macrofauna collection, and one microbial study).

Additional comments: page 3 line 10: at what water depths? Maybe you can mention a range here and see more details in results.

page 3 lines 10-15: Clarify what core was used for what analysis. Right now 4 cores are mentioned but 7 analyses, and the authors end the sentence with "respectively". Were all cores used for everything? Or how was these analyses divided among the cores? How many replicates per analysis?

page 4: How was DNA extracted from sediment and chironomid larval tubes?

page 4 lines 11-14: The author state here that methane is a food source for the studied macrofauna. But considering that methane (i.e. the gas) is not a real food source for these animals, how can this model predict CH4 contribution to their diet? The authors do not know the 13C isotopic compositional values of the methane derived food (i.e. methanotrophs and methanogens.)

page 5 lines 1-5: briefly write how, and with what software, the bioinformatic analyses were conducted.

page 5 lines 1-5: How and with what instrument was the DNA sequenced?

page 5 line 3: Han et al. 2020 is missing in the reference list.

page 5 line 15-16: It would be useful if that was mention earlier, i.e. which stations are oxic or hypoxic and what were the O2 concentrations measured at each station?

page 6 lines 10-27: What were the 13C isotopic composition values for methanotrophs and methanogens? How can the authors know if the Macrofauna ingest or assimilate such methane derived carbon without knowing the values for these food sources?

page 8 lines 29-34: This is aims and I think it is redundant to repeat this in the discussion

page 8 line 40 - page 9 line 1: Oligochaetes is mentioned twice here, is it a mistake?

page 9 lines 5-6: How can the authors be certain that 12% of the contributed carbon is methane derived? Any variation or differences in the 13C isotope values (Fig. 4) might come from other unexplored food sources?

page 10 lines 14-16: This is the first time the radionuclide data is presented in the manuscript. This is results or should be cited if it's already published.

page 12 line 19: Are these previous findings as stated in the sentence? The supplementary data cited indicate that this is results from the current manuscript.

Figure 1: It would be useful to mention in the caption how the degree of eutrophication was defined.

Figure 1: How many cores per station? Are the error bars based on 3 or 9 data points? (i.e. 3 stations or 9 cores with 3 per station)?

Figure 3: Somewhere in the caption it needs to be mentioned that the pie charts show %.

Figure 4: Mention how many data points for each variable.

Figure 5. The authors present results from Han et al. (2020) in the figure. I think this data doesn't belong in this manuscript and can instead be discussed in relation to the results the authors present.

Figure 6 and 7: Are these figures based in all data from all lakes and sediment depths?

Tables 1 and 2: can be moved to supplementary information

Table 4: this is a bit confusing, why are two tests greater and one test less? Perhaps the authors can report the p-values in the results when this data is presented.
* * *

---

## Author Comment (AC1) · 28 Jan 2021

We numbered the comments of Referee 1 for better navigation through the text.

General: In the submitted manuscript by Annika Fiskal et al. the authors sampled sediments from lakes with different levels of eutrophication. The aim was to investigate differences in macrofauna (oligochaetes and chironomids), microbial communities in the sediment as well as on/in the macrofauna, and the contribution of methane derived carbon for macrofauna ingestion/assimilation. The authors found that methane derived carbon is a minor carbon source for macrofauna, and that macrofauna associated prokaryotes are different from sediment prokaryotes.

(1) The conclusions drawn from the stable carbon isotope data is rather uncertain.

This data was used to investigate methane derived carbon and if macrofauna ingest or assimilate this in their bodies. The authors do not know the isotope compositional values for potential food sources derived from methane (such as methane-oxidizing bacteria). From what I have read online methanotroph lipids can range between -45‰ to -65‰ d13C values which is very different from the macrofauna values presented in the submitted manuscript. It would therefore be good if the authors tone down the discussion and conclusions from these findings. The authors can instead focus more on the qPCR data that indicate methane cycling bacteria in the gut of the studied macrofauna. And the isotope data might then be used as supportive data to help support the findings that methane derived carbon is a minor food source.

Author reply: The authors thank the anonymous reviewer for his suggestions. However, we disagree that the interpretation of the isotopic data in relation to methane is uncertain. Numerous studies have published isotopic fractionations during the assimilation of methane by aerobic methane-oxidizing bacteria (we cite three key papers by Krüger et al. (2002),Templeton et al. (2006), and Kankaala et al. (2007), see Table 1 legend). Typically the biomass of methane-oxidizing bacteria on a pure methane diet is depleted in 13C relative to methane by a factor of -30 to -39 per mil. This value is lower, if methane-oxidizing bacteria have additional carbon sources, however, even then their biomass tends to be depleted in 13C relative to methane (e.g., Summons et al. (1994). To account for the uncertainty in the 13C-isotopic compositions of methane-oxidizing bacteria, we calculate their contribution to macrofaunal diet under two end member scenarios: (a) methane-oxidizing bacterial biomass has the isotopic composition of methane (highly conservative), and (b) methane oxidizing bacterial biomass has 13C-isotopic compositions that are -39 more negative than those of methane. Under both scenarios the contribution of methane to the diet of macrofauna through the assimilation of methane-oxidizing bacteria is minor (at most 11.8 per mil under the conservative scenario). We would like to furthermore point out that it is well-established that lacustrine sedimentary macrofauna can acquire isotopic values that indicate a significant contribution of methane-derived carbon by grazing on aerobic methane-oxidizing bac-

teria (also see text, p. 2, L. 31-38).

(2) The discussion is quite long and I think this can be shortened by almost half. The authors go into specific details about the microbiology data on ZOTU level, and paragraphs that mention previous studies with similar results can be shortened. I think the discussion can be better summarized and more focused in relevance to the aim of the study.

Author reply: We will shorten and streamline the Discussion. However, we want to make sure that the many novel findings of our study remain clearly stated. This includes stating the most important ZOTU trends, whenever it is relevant for the interpretation of macrofaunal food sources and trophic levels (also see our replies to your later comments).

(3) I also think that the focus on the macrofauna associated bacteria can be shortened in the manuscript, as is the case in the Abstract where it is just mentioned briefly at the end, while a large part of the discussion is dedicated to this subject.

Author reply: We will aim to strike a more adequate balance between the length of that part in the discussion compared to the abstract. The findings on macrofauna-associated microorganisms are highly novel, provide an important indication of macrofaunal food sources, and even raise the possibility of mutualistic symbioses (also see our reply to comment 17). Therefore they belong into this manuscript. However, we will aim to condense the Discussion and strike a better balance between the Discussion and the Abstract.

(4) There is also essential information missing in the methods such as DNA extraction from the sediment, bioinformatics, and DNA sequencing. It seems that this part is instead presented in a manuscript that is in press (Han et al 2020), however there is no need to present this data as results for this manuscript. I also think it might be misleading to do so and the authors better double-check the journal guidelines for what is acceptable. Instead the authors can mention relevant findings from Han et al. (2020)

in the discussion. If the results are first presented in Han et al. (2020) then it should not be presented again as new results for this manuscript. Furthermore, Han et al. (2020) is missing in the reference list so there is no way for the reviewers to read these methods or results.

Author reply: We thank the anonymous reviewer for this comment and are sorry that this important reference was missing. We have fixed this issue. To be clear: all sequencing data from tubes and macrofauna, and all functional gene data, are new to this study. We only include a small subset of published background sediment 16S rRNA gene from Han et al. (2020) for comparison. These background sediment samples were extracted and sequenced using the same method that we applied in this study. We have tried to make it more clear in the captions of Figures 5 and 7 that only the 'Sediment' sequences were previously published.

(5) I think the authors have a large and interesting dataset and it should definitely be published here or somewhere else. My opinion is that the manuscript needs to be more streamlined and focused on a single story (now it feels like two stories: one geochemical with macrofauna collection, and one microbial study).

Author reply: Thank you for your positive assessment, however, we respectfully disagree. The microbiological part is directly connected to the geochemical and macrofaunal data and provides support of the geochemical and isotopic interpretation (also see reply to comment 17). We, however, understand based on this reviewer comment that it is very important for a more coherent manuscript to make the links between these three data sets more clear.

Additional comments:

(6) page 3 line 10: at what water depths? Maybe you can mention a range here and see more details in results.

Author reply: The water depths are stated in (Fiskal et al., 2019), but we will also

include them here.

(7) page 3 lines 10-15: Clarify what core was used for what analysis. Right now 4 cores are mentioned but 7 analyses, and the authors end the sentence with "respectively". Were all cores used for everything? Or how was these analyses divided among the cores? How many replicates per analysis?

Author reply: We will provide more general information, and refer to the more detailed descriptions in Fiskal et al. (2019). For your information: We only analyzed one sample per sample depth, however, the sampling resolution was high ($\sim$20 depths per core for all DNA, porewater geochemical, and gas analyses). All microsensor measurements were run in triplicates.

(8) page 4: How was DNA extracted from sediment and chironomid larval tubes?

Author reply: The sediment DNA extraction procedure is based on the modular method of Lever et al. (2015) and is described in Han et al., 2020. The same protocol was used for larval tubes. We will state this more clearly.

(9) page 4 lines 11-14: The author state here that methane is a food source for the studied macrofauna. But considering that methane (i.e. the gas) is not a real food source for these animals, how can this model predict CH4 contribution to their diet? The authors do not know the 13C isotopic compositional values of the methane derived food (i.e. methanotrophs and methanogens.)

Author reply: Thank you for catching this. We will change this to carbon source. Regarding the other comment, please see our reply to comment (1).

(10) page 5 lines 1-5: briefly write how, and with what software the bioinformatic analyses were conducted.

Author reply: Thank you for this comment, we will add the missing information to the manuscript.

(11) page 5 lines 1-5: How and with what instrument was the DNA sequenced?

Author reply: Thank you for this comment, we will add the missing information to the manuscript.

(12) page 5 line 3: Han et al. 2020 is missing in the reference list.

Author reply: Thank you. We will fix this.

(13) page 5 line 15-16: It would be useful if that was mention earlier, i.e. which stations are oxic or hypoxic and what were the O2 concentrations measured at each station?

Author reply: Thank you for this comment. We will add this information to the sampling sites and sampling descriptions.

(14) page 6 lines 10-27: What were the 13C isotopic composition values for methanotrophs and methanogens? How can the authors know if the Macrofauna ingest or assimilate such methane derived carbon without knowing the values for these food sources?

Author reply: Please see reply to comment (1) regarding aerobic methanotrophs. Anaerobic methanotrophs and methanogens were present in extremely low numbers in fauna or tubes (see p. 9, L. 15-18). Based on these very low numbers, ingestion of anaerobic methanotrophs and methanogens would only have a minimal impact on the 13C-isotopic compositions of fauna.

(15) page 8 lines 29-34: This is aims and I think it is redundant to repeat this in the discussion

Author reply: Thank you for this comment we will shorten that part in order to keep redundancy to a minimum.

(16) page 8 line 40 - page 9 line 1: Oligochaetes is mentioned twice here, is it a mistake?

Author reply: Yes this is a mistake, we are sorry and will correct this. The correct sentence will appear in the manuscript as follows: "While chironomid communities vary strongly with water depth in the same lakes, oligochaete communities are more uniform across different locations within the same lake."

(17) page 9 lines 5-6: How can the authors be certain that 12% of the contributed carbon is methane derived? Any variation or differences in the 13C isotope values (Fig. 4) might come from other unexplored food sources?

Author reply: Thank you for your question. We cannot be certain what the food sources are based on our own data, but there is a large body of literature on the food sources of chironomid larvae and oligochaetes, which we include in our analyses (for overview see Supplementary Table S4). These studies suggest that both macrofaunal groups have primarily detritus-based food sources (detritus itself, heterotrophic bacteria, primary consumers of heterotrophic bacteria), and/or methane-derived food sources (methane-oxidizing bacteria). In recent years, several studies have suggested a shift from primarily detritus-based food sources to methane-derived food sources ("methane-derived carbon") with increasing trophic state. It has been argued that this is mainly due to the increase in sediment methane production in response to eutrophication, and the resulting shallowing of the methanogenesis zone to layers that are inhabited by sediment macrofauna (e.g., Hershey et al. (2006), Jones and Grey (2011).

We investigated whether such a shift from detritus-derived to methane-derived carbon occurs across the five lakes studied by comparing the d13C-isotopic compositions of detritus (total organic carbon) and methane to those of macrofaunal biomass. Our data indicate a clear and consistent pattern, namely that detritus-derived carbon is the main carbon source of sediment macrofauna. This interpretation is confirmed by analyses of isotopic compositions of dissolved organic carbon and phytoplankton, which are close to those of total organic carbon (Supplementary Table S2 and Figure S2). The high similarity of isotopic values of phytoplankton, total and dissolved organic carbon was expected given that phytoplankton is the main source of detritus (total organic

carbon) in these lakes (see, e.g. , Han et al. (2020), and detritus is the main source of dissolved organic carbon. While methane-derived carbon increases as a carbon source with increasing trophic state similar to previous studies, it is – unlike several of these studies - only a minor carbon source even in the highly eutrophic Lake Baldegg and Lake Greifen (also see answer to Comment 1).

We used a two-end member mixing model to constrain the relative contributions of detritus (TOC) and methane to the biomass-carbon of macrofauna. This is a standard approach for similar two-end member scenarios.

The reviewer is correct that it would in theory be possible for other types of bacteria than aerobic methanotrophs to contribute isotopically light carbon to the biomass of macrofauna. Key examples are methanogens, anaerobic methanotrophs, acetogens, and certain sulfate reducers. However, this is where the tremendous value of our DNA analyses becomes evident. Based on our quantitative DNA analyses and DNA sequence analyses, and based on current knowledge on the dominant groups of bacteria found in "our" sediments, tubes, and fauna, we can rule out that these groups are quantitatively important. Instead, our microbial DNA analyses clearly point to aerobic organoheterotrophs and especially fermentative bacteria being the dominant microorganisms in sediments, tubes, and within macrofauna. These bacteria only minimally fractionate organic carbon (typically 0 to +/-2 per mil relative to the source organic matter), and thus carry the C-isotopic signature of detritus. Consequently, the DNA data nicely support our interpretation of isotopic data, which is that detritus (and most probably detrital-feeding bacteria) is the main carbon source of the lake sedimentary macrofauna – independent of trophic state.

(18) page 10 lines 14-16: This is the first time the radionuclide data is presented in the manuscript. This is results or should be cited if it's already published.

Author reply: Thank you for pointing this out. We will mention the radionuclide measurements in the Materials & Methods and refer to the results of these analyses in the
Results section. We will furthermore carefully consider the possibility of showing the very clear Figure S6 in the main Results section.

(19) page 12 line 19: Are these previous findings as stated in the sentence? The supplementary data cited indicate that this is results from the current manuscript.

Author reply: Thank you for this comment. We are referring to the phylogenetic tree in Fig. S8A. This tree shows the IDs and source environments of the closest related environmental DNA sequences in black. The sequences from our study are the ones that are shown in magenta. These are the sequences we detected in this study. We will change the text to make this more clear, and remove mention of Table S6, since it is not necessary to cite it here. We also realize that the figure caption does not explicitly state which sequences are from this study, and will fix this.

(20) Figure 1: It would be useful to mention in the caption how the degree of eutrophication was defined.

Author reply: Thank you, we will add this information to the figure caption. The degree of eutrophication is based on water column phosphorous concentrations and determined by the Swiss Federal Office of the Environment (BAFU, 2016a, c, b).

(21) Figure 1: How many cores per station? Are the error bars based on 3 or 9 data points? (i.e. 3 stations or 9 cores with 3 per station)?

Author reply: Only 1 core per station so, 3 data points but depth integrated (many depths were sampled and counted per core (10 to 15). We will clarify this in the caption.

(22) Figure 3: Somewhere in the caption it needs to be mentioned that the pie charts show %.

Author reply: Thank you, we will add this information to the caption.

(23) Figure 4: Mention how many data points for each variable.

Author reply: Thank you. We will do this.

(24) Figure 5. The authors present results from Han et al. (2020) in the figure. I think this data doesn't belong in this manuscript and can instead be discussed in relation to the results the authors present.

Author reply: These data are needed for comparison. Also see Author Reply to Comment 4.

(25) Figure 6 and 7: Are these figures based in all data from all lakes and sediment depths?

Author reply: No, they are from a representative number of lake samples and sample depths from each sample category.

(26) Tables 1 and 2: can be moved to supplementary information

Author reply: Thank you, we will do this.

(27) Table 4: this is a bit confusing, why are two tests greater and one test less? Perhaps the authors can report the p-values in the results when this data is presented.

Author reply: Thank you for your comment. We will add the p-value ranges to the table and remove the statement "greater" or "less", which seems to be a source of confusion rather than clarity. We will simply state in the caption that we used (more conservative) one-sided rather than two-sided tests.

References

BAFU, B. f. U.: Der Baldeggersee - Zustand bezüglich Wasserqualität, 2016a.

BAFU, B. f. U.: Der Vierwaldstättersee - Zustand bezüglich Wasserqualität, 2016b.

BAFU, B. f. U.: Der Zürichsee - Zustand bezüglich Wasserqualität, 2016c.

Fiskal, A., Deng, L., Michel, A., Eickenbusch, P., Han, X., Lagostina, L., Zhu, R., Sander, M., Schroth, M., Bernasconi, S., Dubois, N., and Lever, M.: Effects of eutrophication on sedimentary organic carbon cycling in five temperate lakes, Biogeosciences,

16, 3725-3746, 2019.

Han, X. G., Schubert, C. J., Fiskal, A., Dubois, N., and Lever, M. A.: Eutrophication as a driver of microbial community structure in lake sediments, Environ Microbiol, 22, 3446-3462, 2020.

Hershey, A. E., Beaty, S., Fortino, K., Kelly, S., Keyse, M., Luecke, C., O'brien, W., and Whalen, S.: Stable isotope signatures of benthic invertebrates in arctic lakes indicate limited coupling to pelagic production, Limnol Oceanogr, 51, 177-188, 2006.

Jones, R. I. and Grey, J.: Biogenic methane in freshwater food webs, Freshwater Biol, 56, 213-229, 2011.

Kankaala, P., Taipale, S., Nykanen, H., and Jones, R. I.: Oxidation, efflux, and isotopic fractionation of methane during autumnal turnover in a polyhumic, boreal lake, J Geophys Res-Biogeo, 112, 2007.

Krüger, M., Eller, G., Conrad, R., and Frenzel, P.: Seasonal variation in pathways of CH4 production and in CH4 oxidation in rice fields determined by stable carbon isotopes and specific inhibitors, Global Change Biol, 8, 265-280, 2002.

Summons, R. E., Jahnke, L. L., and Roksandic, Z.: Carbon isotopic fractionation in lipids from methanotrophic bacteria: relevance for interpretation of the geochemical record of biomarkers, Geochim Cosmochim Ac, 58, 2853-2863, 1994.

Templeton, A. S., Chu, K. H., Alvarez-Cohen, L., and Conrad, M. E.: Variable carbon isotope fractionation expressed by aerobic CH4-oxidizing bacteria, Geochim Cosmochim Ac, 70, 1739-1752, 2006.

---

## Referee Comment (RC2) · Anonymous Referee #2 · 26 Mar 2021

The manuscript submitted by Annika Fiskal et al. aimed to investigate various carbon sources' contribution to the benthic macrofaunal biomass across the sediments from five lakes in the temperate region. Though the introduction is short, the section is well written with current knowledge and associated gap addressed through the present work. The methodology is well described and elaborated. The results and discussion section are well written, along with all the pertinent figures and tables. The authors have substantially concluded the paper. The present study deals with methane-derived carbon to the benthic macrofaunal community, a poorly studied area that will give additional understating to the benthic carbon cycle. Therefore the communicated manuscript is recommended for acceptance with few minor technical revisions. Comments mentioned below may be considered while revising the MS.

Materials and methods: It has been referred to as Fiskal et al. 2019 about the sampling locations and map in the method section. A map and short description of the depths would be catchier to easy access for the readers because few hypoxic depths are present too. How many replicates were collected for estimation of the density and biomass of macrofauna? As per the reference mentioned for detail collection in Fiskal et al. 2019, it appeared that only a single core at each station had been considered for macrofaunal estimation. What is the justification for single-core collection for macrofaunal quantification? It is always suggested to collect sufficient replicates to estimate the benthic faunal community and be statistically justified because macrofauna quantification could impact estimating the budget of other related data. p.5. L 4-5. The PCoA analysis line may be added to the statistical analyses section.

Results: Page 5, lines17 − 19, expressing of density may be like average density 75±86 ind.m-2. It should be mentioned that SD/SE is used to expressing the density data.

---

## Author Comment (AC2) · 29 Mar 2021

We numbered the Referee comments for better navigation through the text

The manuscript submitted by Annika Fiskal et al. aimed to investigate various carbon sources' contribution to the benthic macrofaunal biomass across the sediments from five lakes in the temperate region.

(1) Though the introduction is short, the section is well written with current knowledge and associated gap addressed through the present work. The methodology is well described and elaborated. The results and discussion section are well written, along with all the pertinent figures and tables. The authors have substantially concluded the

paper. The present study deals with methane-derived carbon to the benthic macro-faunal community, a poorly studied area that will give additional understating to the benthic carbon cycle. Therefore the communicated manuscript is recommended for acceptance with few minor technical revisions. Comments mentioned below may be considered while revising the MS.

Author reply: We very much appreciate this positive feedback from the referee.

(2) Discussion paper Materials and methods: It has been referred to as Fiskal et al. 2019 about the sampling locations and map in the method section. A map and short description of the depths would be catchier to easy access for the readers because few hypoxic depths are present too.

Author reply: thank you very much for your comment, we will add the map and table with the station information to the ms.

(3) How many replicates were collected for estimation of the density and biomass of macrofauna? As per the reference mentioned for detail collection in Fiskal et al. 2019, it appeared that only a single core at each station had been considered for macrofaunal estimation. What is the justification for single-core collection for macrofaunal quantification? It is always suggested to collect sufficient replicates to estimate the benthic faunal community and be statistically justified because macrofauna quantification could impact estimating the budget of other related data.

Author reply: Thank you very much for this comment. It is true that only one core was analyzed for macrofaunal abundance at each of the three stations per lake. Nevertheless, we are confident that these abundances are representative for the following reasons:

(a) The cores have large diameters (15 cm), also in relation to the size of the organisms, and thus cover a large area of the sediment surface.

(b) The observed trends in tubificids and chironomids follow clear lake-specific trends,

both in composition and abundance, suggesting that the depth of sampling was sufficient to address the questions investigated (also see Figure 1, Figure 2, and Table S5).

(c) We revisited one station in each eutrophic Lake Baldegg and oligotrophic Lake Lucerne at different times of the year (early to late autumn, as well as mid-spring to early summer) during 3 different years. Oligochaete abundances in 2016, 2017 and 2019 in Lake Baldegg at 66 m water depth showed only minor interannual variations ($\sim$8450$\pm$665 ind. m-2). The same was true for chironomid larvae in Lake Lucerne at 24 m water depth ($\sim$1036$\pm$62 ind. m-2).

This suggests that the abundances presented here are indeed a good representation even if only one core per station was sampled.

(4) p.5. L 4-5. The PCoA analysis line may be added to the statistical analyses section.

Author reply: Thank you for this comment, we agree and will add this.

(5) Results: Page 5, lines17 – 19, expressing of density may be like average density 75$\pm$86 ind.m-2. It should be mentioned that SD/SE is used to expressing the density data.

Author reply: Thank you very much for this comment. These values refer to standard deviations. We will state this in the text.

---

## Author Response (AR1)

*Point by point response to the referee comments*

We numbered the Referee comments for better navigation through the text.

*Answers are in blue*

*Referee 1*

*General:*
*In the submitted manuscript by Annika Fiskal et al. the authors sampled sediments from lakes with different levels of eutrophication. The aim was to investigate differences in macrofauna (oligochaetes and chironomids), microbial communities in the sediment as well as on/in the macrofauna, and the contribution of methane derived carbon for macrofauna ingestion/assimilation. The authors found that methane derived carbon is a minor carbon source for macrofauna, and that macrofauna associated prokaryotes are different from sediment prokaryotes.*

*(1) The conclusions drawn from the stable carbon isotope data is rather uncertain. This data was used to investigate methane derived carbon and if macrofauna ingest or assimilate this in their bodies. The authors do not know the isotope compositional values for potential food sources derived from methane (such as methane-oxidizing bacteria). From what I have read online methanotroph lipids can range between -45‰ to -65‰ d13C values which is very different from the macrofauna values presented in the submitted manuscript. It would therefore be good if the authors tone down the discussion and conclusions from these findings. The authors can instead focus more on the qPCR data that indicate methane cycling bacteria in the gut of the studied macrofauna. And the isotope data might then be used as supportive data to help support the findings that methane derived carbon is a minor food source.*

**Author reply:**

The authors thank the anonymous reviewer for his suggestions. However, we disagree that the interpretation of the isotopic data in relation to methane is uncertain. Numerous studies have published isotopic fractionations during the assimilation of methane by aerobic methane-oxidizing bacteria (we cite three key papers by Krüger et al. (2002),Templeton et al. (2006), and Kankaala et al. (2007), see Table **2** legend). Typically the biomass of methane-oxidizing bacteria on a pure methane diet is depleted in $^{13}C$ relative to methane by a factor of -30 to -39 per mil. This value is lower, if methane-oxidizing bacteria have additional carbon sources, however, even then their biomass tends to be depleted in $^{13}C$ relative to methane (e.g., Summons et al. (1994)).

To account for the uncertainty in the $^{13}C$-isotopic compositions of methane-oxidizing bacteria, we calculate their contribution to macrofaunal diet under two end member scenarios: (a) methane-oxidizing bacterial biomass has the same isotopic composition as methane (highly conservative), and (b) methane oxidizing bacterial biomass has $^{13}C$-isotopic compositions that are -39 more negative than those of methane. Under both scenarios the contribution of methane to the diet of macrofauna through the assimilation of methane-oxidizing bacteria is minor (at most 11.8 per mil under the conservative scenario).

We would like to furthermore point out that it is well-established that lacustrine sedimentary macrofauna can acquire isotopic values that indicate a significant contribution of methane-derived carbon by grazing on aerobic methane-oxidizing bacteria (also see text, p. 2, L. 31-38).

*(2) The discussion is quite long and I think this can be shortened by almost half. The authors go into specific details about the microbiology data on ZOTU level, and paragraphs that mention previous studies with similar results can be shortened. I think the discussion can be better summarized and more focused in relevance to the aim of the study.*

**Author reply:**

We have shortened and streamlined the Discussion by over 400 words (from 3,446 to 3,013 words; heavily edited parts are in blue), without dismissing the aims of our study, which, as stated in the end of our Introduction, include microbiology (p. 2, l. 42 to p. l. 7). The Discussion is now slightly over 4 pages long, which is not particularly long for a full-length interdisciplinary article like this.

Discussions at the ZOTU-level are important for the analyses of symbionts, as most macrofaunal specimens are dominated (>50% of sequence reads) by single ZOTUs that are rare in the microbially much more diverse surrounding sediment. Such high read percentages of single ZOTUs are extremely rare in nature, and are clear indications of symbiotic relationships. Given the focus of this study on the carbon sources of sedimentary macrofauna, it is important to note and discuss the potential dietary role of these symbionts.

*(3)  I also think that the focus on the macrofauna associated bacteria can be shortened in the manuscript, as is the case in the Abstract where it is just mentioned briefly at the end, while a large part of the discussion is dedicated to this subject.*

**Author reply:**

Thank you. We have followed this advice and aimed to strike a more adequate balance between the microbiology part (1,183 words) and other parts of the discussion (1,830 words).

We would like to point, however, that nearly half of the Abstract (101/223 words) is on microbiology. Moreover, the findings on macrofauna-associated microorganisms are highly novel, provide important insights into macrofaunal food sources, and even raise the possibility of previously unknown mutualistic symbioses (also see replies to comments 2 and 17). Therefore we consider it very important to discuss the microbiological data in this manuscript.

*(4)  There is also essential information missing in the methods such as DNA extraction from the sediment, bioinformatics, and DNA sequencing. It seems that this part is instead presented in a manuscript that is in press (Han et al 2020), however there is no need to present this data as results for this manuscript. I also think it might be misleading to do so and the authors better double-check the journal guidelines for what is acceptable. Instead the authors can mention relevant findings from Han et al. (2020) in the discussion. If the results are first presented in Han et al. (2020) then it should not be presented again as new results for this manuscript. Furthermore, Han et al. (2020) is missing in the reference list so there is no way for the reviewers to read these methods or results.*

**Author reply:**

We thank the anonymous reviewer for this comment and are sorry that this important reference was missing. We have fixed this issue.

To be clear: all sequencing data from tubes and macrofauna, and all functional gene data, are new to this study. We only include a small subset of data on sediment 16S rRNA genes from Han et al. (2020) for comparison. These samples were extracted and sequenced using the same method that we applied in this study.

We have tried to make it more clear in the captions of **Figures 6 through 8** that only the 'Sediment' sequences were previously published.

*(5) I think the authors have a large and interesting dataset and it should definitely be published here or somewhere else. My opinion is that the manuscript needs to be more streamlined and focused on a single story (now it feels like two stories: one geochemical with macrofauna collection, and one microbial study).*

**Author reply:**
Thank you for your overall positive assessment, as well as your critical comment regarding the lack of integration of the geochemical and microbiological data. In order to streamline the story more, we have added short text in various places throughout the manuscript that aims to more clearly link the microbiological data to the geochemical data:

P. 5, L6

P. 6, L39-40

P. 8, L5-9, L14-15, L20-25

P. 9, L16-18

*Additional comments:*

*(6) page 3 line 10: at what water depths? Maybe you can mention a range here and see more details in results.*

**Author reply:**
The water depths are stated in (Fiskal et al., 2019), but we will also include them here. **We have added a map indicating the station as well as a table with water depths and oxygen concentration to the main text (Figure 1 and Table 1).**

*(7) page 3 lines 10-15: Clarify what core was used for what analysis. Right now 4 cores are mentioned but 7 analyses, and the authors end the sentence with "respectively". Were all cores used for everything? Or how was these analyses divided among the cores? How many replicates per analysis?*

**Author reply:**
We have revised this text to make it more clear **(p. 3, L12-24).**

*(8) page 4: How was DNA extracted from sediment and chironomid larval tubes?*

**Author reply:**
The sediment DNA extraction procedure is based on the modular method of Lever et al. (2015) and is described in Han et al., 2020. See revised text **(p. 4, L22-26).**

*(9) page 4 lines 11-14: The author state here that methane is a food source for the studied macrofauna. But considering that methane (i.e. the gas) is not a real food source for these animals, how can this model predict CH4 contribution to their diet? The authors do not know the 13C isotopic compositional values of the methane derived food (i.e. methanotrophs and methanogens.)*

**Author reply:**
Thank you for catching this. **We have changed this to carbon sources in the main text (p.4 L14).**

Regarding the other comment, please see our reply to comment (1).

*(10) page 5 lines 1-5: briefly write how, and with what software the bioinformatic analyses were conducted.*

**Author reply:**
Thank you for this comment, we will add the missing information to the manuscript.
**The missing information was added to the material and methods (p.5 L11-13).**

*(11) page 5 lines 1-5: How and with what instrument was the DNA sequenced?*

**Author reply:**
Thank you for this comment, we will add the missing information to the manuscript.
**The missing information was added to the material and methods (p.5, L11).**

*(12) page 5 line 3: Han et al. 2020 is missing in the reference list.*

**Author reply:**
Thank you.
**We have fixed this.**

*(13) page 5 line 15-16: It would be useful if that was mention earlier, i.e. which stations are oxic or hypoxic and what were the O2 concentrations measured at each station?*

**Author reply:**
Thank you for this comment. We will add this information to the sampling sites and sampling descriptions.
**Added in new Table 1.**

*(14) page 6 lines 10-27: What were the 13C isotopic composition values for methanotrophs and methanogens? How can the authors know if the Macrofauna ingest or assimilate such methane derived carbon without knowing the values for these food sources?*

**Author reply:**
Please see reply to comment (1) regarding the carbon isotopic values of aerobic methanotrophs.

Anaerobic methanotrophs and methanogens were present in extremely low numbers in fauna and tubes (**see p. 9, L. 23-31**). Based on these very low numbers, ingestion of anaerobic methanotrophs and methanogens would only have a minimal impact on the $^{13}$C-isotopic compositions of fauna.

*(15) page 8 lines 29-34: This is aims and I think it is redundant to repeat this in the discussion*

**Author reply:**
Thank you for this comment but we respectfully disagree. It is common practice to briefly restate the context and initial goals of a study in the beginning of the Discussion. We consider this a good practice that helps remind readers of the original aims of the study.

*(16) page 8 line 40 - page 9 line 1: Oligochaetes is mentioned twice here, is it a mistake?*

**Author reply:**
Yes this is a mistake, we are sorry and will correct this. The correct sentence will appear in the manuscript as follows: "While **chironomid** communities vary strongly with water depth in the same lakes, oligochaete communities are more uniform across different locations within the same lake." **p.8 L36**

*(17) page 9 lines 5-6: How can the authors be certain that 12% of the contributed carbon is methane derived? Any variation or differences in the 13C isotope values (Fig. 4) might come from other unexplored food sources?*

**Author reply:**
Thank you for your question. We cannot be certain what the food sources are based on our own data, but there is a large body of literature on the food sources of chironomid larvae and oligochaetes, which we include in our analyses (for overview see Supplementary Table S4). These studies suggest that both macrofaunal groups have primarily detritus-based food sources (detritus itself, heterotrophic bacteria, primary consumers of heterotrophic bacteria), and/or methane-derived food sources (methane-oxidizing bacteria). In recent years, several studies have suggested a shift from primarily detritus-based food sources to methane-derived food sources ("methane-derived carbon") with increasing trophic state. It has been argued that this is mainly due to the increase in sediment methane production in response to eutrophication, and the resulting shallowing of the methanogenesis zone to layers that are inhabited by both aerobic methanotrophs and sediment macrofauna (e.g., Hershey et al. (2006), Jones and Grey (2011).

We investigated whether such a shift from detritus-derived to methane-derived carbon occurs across the five lakes studied by comparing the $\delta^{13}$C-isotopic compositions of detritus (total organic carbon) and methane to those of macrofaunal biomass. Our data indicate a clear and consistent pattern, namely that detritus-derived carbon is the main carbon source of sediment macrofauna. This interpretation is confirmed by analyses of isotopic compositions of dissolved organic carbon and phytoplankton, which are close to those of total organic carbon (Supplementary Table S2 and Figure S2). The high similarity of isotopic values of phytoplankton, total and dissolved organic carbon was expected given that phytoplankton is the main source of detritus (total organic carbon) in these lakes (see, e.g. , Han et al. (2020)), and detritus is the main source of dissolved organic carbon. While methane-derived carbon increases as a carbon source with increasing trophic state similar to previous studies, it is – unlike several of these studies - only a minor carbon source even in the highly eutrophic Lake Baldegg and Lake Greifen (also see answer to Comment 1).

We used a two-end member mixing model to constrain the relative contributions of detritus (TOC) and methane to the biomass-carbon of macrofauna. This is a standard approach for similar two-end member scenarios. We end up with a maximum contribution of methane-derived carbon of 6-12% (Lake Zug and Lake Baldegg; Table 2). Hereby 6% corresponds to the most conservative estimate (assumes methanotrophic bacteria have -39 per mil lower $^{13}$C-values than CH4), and 12% corresponds to the maximum number (assumes methanotrophic bacteria have 13C biomass compositions equivalent to those of methane).

The reviewer is correct that it would in theory be possible for other types of microorganisms besides aerobic methanotrophs to contribute isotopically light carbon to the biomass of macrofauna. Key examples are methanogens, anaerobic methanotrophs, acetogens, and certain sulfate reducers. However, this is where our DNA analyses are important. Our quantitative DNA analyses and DNA sequence analyses, and current knowledge on the dominant groups of microorganisms found in "our" sediments, tubes, and fauna, argue against these groups being quantitatively important – thus confirming our conclusion that the macrofauna studied primarily incorporate detrital organic carbon.

In addition, it is possible that tubificids and chironomid larvae mainly feed on isotopically slightly depleted detrital fractions in the eutrophic lakes. This explanation would not require any grazing on microbial populations with isotopically depleted biomass, and we in fact acknowledge this possibility (p. 11, L. 5-7), though we also clearly state why available data from this study does not support it (p. 11, L.7-11).

*(18) page 10 lines 14-16: This is the first time the radionuclide data is presented in the manuscript. This is results or should be cited if it's already published.*

**Author reply:**
Thank you for pointing this out. We will mention the radionuclide measurements in the Materials & Methods and refer to the results of these analyses in the Results section.
**We now mention the radionuclide measurements in the material and methods (p.3 L17) as well as in the results section (p. 5, L22-23). As mentioned in the results text, these data were used but not shown in Fiskal et al. (2019).**

*(19) page 12 line 19: Are these previous findings as stated in the sentence? The supplementary data cited indicate that this is results from the current manuscript.*

**Author reply:**
Thank you for this comment. We are referring to the phylogenetic tree in Fig. S8A. This tree shows the IDs and source environments of the closest related environmental DNA sequences in black. The sequences from our study are the ones that are shown in magenta and have not been published before. We will change the text to make this more clear, and remove mention of Table S6, since it is not necessary to cite it here. We also realize that the figure caption does not explicitly state which sequences are from this study, and will fix this.

**We have changed the figure caption (Figure S8, SI, p19, L1-3) and clarified the main text (p11, L34-36).**

*(20) Figure 1: It would be useful to mention in the caption how the degree of eutrophication was defined.*

**Author reply:**
Thank you, we will add this information to the figure caption. **The degree of eutrophication is based on water column phosphorous concentrations measured by the Swiss Federal Office of the Environment, which uses the OECD model which declares lakes with average values of ≤15 mg P m-3 as oligotrophic, lakes with 15-45 mg P m-3 as mesotrophic, and lakes with >45 mg P m-3 as eutrophic (Vollenweider and Kerekes, 1982). We have added this information and cited the reference in the Figure 2 caption.**

*(21) Figure 1: How many cores per station? Are the error bars based on 3 or 9 data points? (i.e. 3 stations or 9 cores with 3 per station)?*

**Author reply:**
Only 1 core per station so the standard deviation is based on total macrofaunal counts from 3 cores. **We have clarified this in the Figure 2 caption. The fact that one core was sampled for macrofauna at each station is stated in the Materials & Methods (p. 3, L13-14).**

*(22) Figure 3: Somewhere in the caption it needs to be mentioned that the pie charts show %.*
**Author reply:**
Thank you, we will add this information to the caption. **Please see revised Figure 4 caption.**

*(23) Figure 4: Mention how many data points for each variable.*

**Author reply:**
Thank you. We **have done** this.
*(24) Figure 5. The authors present results from Han et al. (2020) in the figure. I think this data doesn't belong in this manuscript and can instead be discussed in relation to the results the authors present.*

**Author reply:**
The 16S rRNA gene sequence data from Han et al. (2020) are needed for comparison to the new 16S rRNA gene sequence data on chironomid larvae, chironomid larval tubes, and oligochaetes. By comparing them, we can show that bacterial communities are highly similar between sediments and larval tubes, but very different between larval tubes and both groups of macrofauna. This is an important finding.

*(25) Figure 6 and 7: Are these figures based in all data from all lakes and sediment depths?*

**Author reply:**
No, specimens of chironomid larvae, chironomid larval tubes, and oligochaetes were selected from all stations where they were found, covering the entire depth ranges at which they were present, and then compared to a subset of sediment samples from the same stations and depths. All information on station and depth of origin is included in the bar chart in Fig. 6. The trends and differences between sample types are very clear (also see Fig. 7).

*(26) Tables 1 and 2: can be moved to supplementary information*

**Author reply:**
Thank you, we **have done** this.

*(27) Table 4: this is a bit confusing, why are two tests greater and one test less? Perhaps the authors can report the p-values in the results when this data is presented.*

**Author reply:**
Thank you for your comment. We will add the p-value ranges to the table and remove the statement "greater" or "less", which seems to be a source of confusion rather than clarity. We will simply state in the caption that we used (more conservative) one-sided rather than two-sided tests.

**References**

Fiskal, A., Deng, L., Michel, A., Eickenbusch, P., Han, X., Lagostina, L., Zhu, R., Sander, M., Schroth, M., Bernasconi, S., Dubois, N., and Lever, M.: Effects of eutrophication on sedimentary organic carbon cycling in five temperate lakes, Biogeosciences, 16, 3725-3746, 2019.

Han, X. G., Schubert, C. J., Fiskal, A., Dubois, N., and Lever, M. A.: Eutrophication as a driver of microbial community structure in lake sediments, Environ Microbiol, 22, 3446-3462, 2020.

Hershey, A. E., Beaty, S., Fortino, K., Kelly, S., Keyse, M., Luecke, C., O'brien, W., and Whalen, S.: Stable isotope signatures of benthic invertebrates in arctic lakes indicate limited coupling to pelagic production, Limnol Oceanogr, 51, 177-188, 2006.

Jones, R. I. and Grey, J.: Biogenic methane in freshwater food webs, Freshwater Biol, 56, 213-229, 2011.

Kankaala, P., Taipale, S., Nykanen, H., and Jones, R. I.: Oxidation, efflux, and isotopic fractionation of methane during autumnal turnover in a polyhumic, boreal lake, J Geophys Res-Biogeo, 112, 2007.

Krüger, M., Eller, G., Conrad, R., and Frenzel, P.: Seasonal variation in pathways of $CH_4$ production and in $CH_4$ oxidation in rice fields determined by stable carbon isotopes and specific inhibitors, Global Change Biol, 8, 265-280, 2002.

Summons, R. E., Jahnke, L. L., and Roksandic, Z.: Carbon isotopic fractionation in lipids from methanotrophic bacteria: relevance for interpretation of the geochemical record of biomarkers, Geochim Cosmochim Ac, 58, 2853-2863, 1994.

Templeton, A. S., Chu, K. H., Alvarez-Cohen, L., and Conrad, M. E.: Variable carbon isotope fractionation expressed by aerobic $CH_4$-oxidizing bacteria, Geochim Cosmochim Ac, 70, 1739-1752, 2006.

Vollenweider, R. and Kerekes, J.: Eutrophication of Waters: Monitoring, Assessment and Control. OECD, Paris, 154, 1982.

*Referee 2*

The manuscript submitted by Annika Fiskal et al. aimed to investigate various carbon sources' contribution to the benthic macrofaunal biomass across the sediments from five lakes in the temperate region.

(1) Though the introduction is short, the section is well written with current knowledge and associated gap addressed through the present work. The methodology is well described and elaborated. The results and discussion section are well written, along with all the pertinent figures and tables. The authors have substantially concluded the paper. The present study deals with methane-derived carbon to the benthic macrofaunal community, a poorly studied area that will give additional understating to the benthic carbon cycle. Therefore the communicated manuscript is recommended for acceptance with few minor technical revisions. Comments mentioned below may be considered while revising the MS.

Author reply: We very much appreciate this positive feedback by the referee.

(2) Discussion paper Materials and methods: It has been referred to as Fiskal et al. 2019 about the sampling locations and map in the method section. A map and short description of the depths would be catchier to easy access for the readers because few hypoxic depths are present too.

Author reply: thank you very much for your comment, we will add the map and table with the station information to the ms.

**We have now included the map and station table (new Figure 1 and Table 1) in the main text.**

(3) How many replicates were collected for estimation of the density and biomass of macrofauna? As per the reference mentioned for detail collection in Fiskal et al. 2019, it appeared that only a single core at each station had been considered for macrofaunal estimation. What is the justification for single-core collection for macrofaunal quantification? It is always suggested to collect sufficient replicates to estimate the benthic faunal community and be statistically justified because macrofauna quantification could impact estimating the budget of other related data.

Author reply: Thank you very much for this comment. It is true that only one core was analyzed for macrofaunal abundance at each of the three stations per lake. Nevertheless, we are confident that these abundances are representative for the following reasons:

(1) The cores have large diameters (15 cm), also in relation to the size of the organisms, and thus cover a large area of the sediment surface.
(2) The observed trends in tubificids and chironomids follow very clear lake-specific trends, both in composition and abundance, suggesting that the degree of replication was sufficient to address the questions investigated (also see Figure 1, Figure 2, and Table S5).
(3) We revisited one station in eutrophic Lake Baldegg and one station in oligotrophic Lake Lucerne at different times of the year (early to late autumn, as well as mid spring to early summer) during 3 different years. Oligochaete abundances in 2016, 2017 and 2019 in Lake Baldegg at 66 m water depth showed only minor variations between sampling dates (~8450±665 ind. m$^{-2}$). The same was true for chironomid larvae in Lake Lucerne at 24 m water depth (~1036±62 ind. m$^{-2}$). These variations in macrofaunal densities within the same stations sampled at different dates are smaller than the variations in macrofaunal densities observed between stations from the same lakes around the same time during the main

sampling campaign in 2016. Consequently, the abundances presented here are representative, even though only one core per station was sampled.

(4) p.5. L 4-5. The PCoA analysis line may be added to the statistical analyses section.

Author reply: Thank you for this comment, we agree and **have added** this (**p.5 18-19).**

(5) Results: Page 5, lines17 – 19, expressing of density may be like average density 75±86 ind.m-2. It should be mentioned that SD/SE is used to expressing the density data.

Author reply: Thank you very much for this comment. These values refer to standard deviations. We will state this in the text., **p.5 L23-27**

---

## Author Response (AR2)

**Referee report 1**

I think the authors have done a good job improving the manuscript and it is now suitable for publication in Biogeosciences. Even though the data is based on single cores collected at various water depths I think the overall conclusions, based on all samples from all stations, holds and makes ecological sense. With both the isotope and microbiology data indicating a low contribution of methane-derived-carbon to the biomass of the studied macrofauna.

I just have a few technical minor comments that can be fixed with a quick resubmission or potentially during the proofing stage.

**Answer: We thank the anonymous referee for the positive feedback and helpful comments.**

MINOR COMMENTS

Page 5, line 2: Clarify that DNA was pooled after library preparation.

**Answer: Thank you we have added this.**

Page 5, line 2: It would be useful to mention the sequencing setup used with the MiSeq to know the fragment length sequenced. If it is the same as Han et al (2020) it should say: paired-end 2 x 300 bp.

**Answer: Thank you we have added this missing information.**

Page 5, line 5: It would be great if it would be possible to write a few sentences summarizing the workflow of the bioinformatics done. I know this is mentioned in Han et al (2020) but considering the subheading says "Bioinformatics analyses" at least something is expected. For example at the end of line 5 there could be two new sentences: Briefly, raw sequence data was initially quality trimmed using seqtk (ref), paired-end reads merged using FLASh (ref). This was followed by a final quality filtering using prinseq (ref), and the sequences were then used to generate ZOTUs with USEARCH unoise3 using a 97% clustering identity (ref).

**Answer: Thank you for the suggestions. We have added a few more sentences as suggested.**

page 8, line 40: Why is this text Bold? It also ends with a parenthesis.

**Answer: Thank you for noticing. We had a technical issue and have fixed this now.**

page 9, line 35: "for" misspelled

**Answer: Thank you for spotting this. We corrected this.**

page 9, line 38: Why is this text Bold? It also ends with a semicolon.
page 0, line 14: Why is this text Bold? It also ends with a parenthesis.

**Answer: Thank you for spotting this. We have again fixed this.**

Table 1: The O2 data is shown as a range without an explanation in the table legend. Is this duplicate samples? Or perhaps a range of values from seasonal sampling?

**Answer: Thank you for noticing. We have clarified this in the table caption. These are indeed oxygen ($O_2$) concentrations over the time course of one year.**

Table 2: This table shows +/- values with no explanation in the table legend if this is standard deviation or standard error, also the number of replicates are not mentioned.

**Answer: Thank you for noticing. We have made the table caption more clear. These are average ± standard deviation values for all macrofauna samples that could be matched with corresponding TOC values from the same sediment depth (± 2cm).**

Table 3: Here it could be useful to clarify which of the tested factors were higher for each test, e.g. in the first cell it could say: ***, Oligochaetes higher, (p=0.000002).

**Answer: Thank you for pointing this out. We have changed this as suggested.**